

# GEM-MACH-PAH (rev2488): a new high-resolution chemical transport model for North American PAHs and benzene

Cynthia H. Whaley[1,2], Elisabeth Galarneau[1], Paul A. Makar[1],
Ayodeji Akingunola[1], Wanmin Gong[1], Sylvie Gravel[1], Michael D. Moran[1],
Craig Stroud[1], Junhua Zhang[1], and Qiong Zheng[1]

[1]Air Quality Research Division, Environment and Climate Change Canada, 4905 Dufferin Street,
Toronto, ON, M3H 5T4, Canada
[2]Climate Research Division, Environment and Climate Change Canada, 4905 Dufferin Street,
Toronto, ON, M3H 5T4, Canada

*Correspondence to:* Cynthia Whaley (cynthia.whaley@canada.ca)

**Abstract.** Environment and Climate Change Canada's online air quality forecasting model, GEM-MACH, was extended to simulate atmospheric concentrations of benzene and seven polycyclic aromatic hydrocarbons (PAHs): phenanthrene, anthracene, fluoranthene, pyrene, benz(a)anthracene, chrysene, and benzo(a)pyrene. In the expanded model, benzene and PAHs are emitted from major

point, area, and mobile sources, with emissions based on recent emission factors. Modelled PAHs undergo gas-particle partitioning (whereas benzene is only in the gas phase), atmospheric transport, oxidation, cloud processing, and dry and wet deposition. To represent PAH gas-particle partitioning, the Dachs-Eisenreich scheme was used, and we have improved gas-particle partitioning parameters based on an empirical analysis to get significantly better gas-particle partitioning results than the

previous North American PAH model, AURAMS-PAH. Other added process parameterizations include the particle phase benzo(a)pyrene reaction with ozone via the Kwamena scheme and gas-phase scavenging of PAHs by snow via vapor sorption to the snow surface.

The resulting GEM-MACH-PAH model was used to generate the first online model simulations of PAH emissions, transport, chemical transformation and deposition for a high resolution domain

(2.5-km grid cell spacing) in North America, centered on the PAH-data-rich region of southern Ontario, Canada and the north-eastern United States. Model output for two seasons was compared to measurements from three monitoring networks spanning Canada and the U.S. Average summertime model results were found to be statistically indistinguishable from measurements of benzene and all seven PAHs. The same was true for the winter seasonal mean, except for benzo(a)pyrene (BaP),

which had a statistically significant positive bias. We present evidence that the benzo(a)pyrene results may be ameliorated via further improvements to PM and oxidant processes and transport. Our analysis focused on four key components to the prediction of atmospheric PAH levels: spatial variability; sensitivity to mobile emissions; gas-particle partitioning; and wet deposition. Spatial variability of

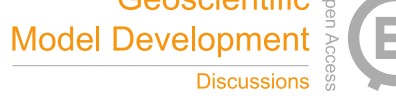



PAHs/PM$_{2.5}$ at 2.5-km resolution was found to be comparable to measurements. Predicted ambient

surface concentrations of benzene and the PAHs were found to be critically dependent on mobile emission factors, indicating the mobile emissions sector has a significant influence on ambient PAH levels in the study region. PAH wet deposition was overestimated due to additive precipitation biases in the model and the measurements. Our overall performance evaluation suggests that GEM-MACH-PAH can provide seasonal estimates for benzene and PAHs and be suitable for emissions scenario

simulations.

## 1 Introduction

Polycyclic aromatic hydrocarbons (PAHs) are semi-volatile atmospheric pollutants that have numerous negative health effects (some are carcinogenic, mutagenic, and teratogenic) (Kim et al., 2013). Measurements of PAHs in North America are sparse in both time (typically 24-hour averages, every 6

days) and space (limited surface measurement networks), yet show ambient concentrations that regularly exceed the Ontario provincial government's health-based threshold (Galarneau et al., 2016). Similarly, benzene is a gas-phase single-ring aromatic hydrocarbon, is a known carcinogen, and also exceeds atmospheric health-based guidelines (Galarneau et al., 2016). Accurate, 3-dimensional modelling of PAHs and benzene can fill in the space-time gaps of the measurements, identify atmo-

spheric processes that are responsible for the threshold exceedances, and simulate effects of emissions scenarios.

AURAMS (A Unified Regional Air quality Modelling System) was an offline (meteorology from a weather forecast model used as an input), Eulerian 3-D chemical transport model (CTM) developed by Environment and Climate Change Canada (ECCC). In Galarneau et al. (2014), AU-

RAMS was modified to include seven PAH species (phenanthrene, anthracene, fluoranthene, pyrene, benz(a)anthracene, chrysene, and benzo(a)pyrene – hereafter abbreviated to PHEN, ANTH, FLRT, PYR BaA, CHRY, BaP, respectively). AURAMS-PAH included emissions, transport, gas-particle partitioning, oxidation of the gas-phase PAHs with OH, dry deposition, and wet deposition of the particle-phase PAHs. This model was able to accurately simulate the 2002 annual average PAH

concentrations in North America when compared to 45 measurement sites, located in Ontario, the north-eastern U.S., and California. However, the AURAMS-PAH gas-particle partitioning overpredicted the gas phase for the lighter PAH species, and was employed at relatively poor time and spatial resolutions. It was also missing two known PAH loss processes: the surface reaction of O$_3$ on particulate BaP (Kwamena et al., 2004, 2007; Ringuet et al., 2012; Keyte et al., 2013; Liu et al.,

2014), and snow scavenging of gas-phase PAHs (Franz and Eisenreich, 1998; Daly and Wania, 2004; Lei and Wania, 2004; Skrdlíková et al., 2011). These missing processes, along with the coarse (42-km) spatial resolution, may have contributed to the differences between model results and measure-



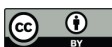

ments. Also, this model used PAH emission factors for mobile emissions which are now out-of-date for representing the modern vehicle fleet.

Other PAH CTMs include GEOS-Chem (Friedman and Selin, 2012; Thackray et al., 2015), which is a global model, CMAQ, which was run on the Europe continental domain in Aulinger et al. (2007), and in North America in Zhang et al. (2016, 2017), FARM (Flexible Air quality Regional Model) (Gariazzo et al., 2007), which is a regional model, applied for the region of Rome, Italy, and WRF-Chem-PAH (Mu et al., 2017), which modelled East Asia. The most relevant of these model studies

to our own is the one by Zhang et al. (2016, 2017), whereby they ran CMAQ with 16 PAH species added, at 36-km resolution in a mainly U.S. domain (that included parts of Canada and Mexico), evaluated their model results against NATTS measurements, and used their results to determine cancer risk to U.S. human populations from various sources.

Therefore, the goal of this study is to update and improve ECCC's PAH modelling capabilities by

using a more advanced model framework, updating emission inventories, and utilizing better process representation of PAHs than were used in AURAMS-PAH to allow better exploration of PAH processes and scenarios. To achieve this goal, GEM-MACH (Global Environment Multiscale model – Modelling Air quality and CHemistry), ECCC's next generation, online air quality forecasting model (meteorology and air-quality are predicted in the same code) was modified to include the same seven

PAH species, as well as benzene. PAH processes parameterization were improved in the following ways: 1. On-road mobile PAH emissions were updated with more recent data, to better represent the modern vehicle fleet; 2. gas-particle partitioning parameters were improved based on empirical results and analysis of AURAMS model output; 3. process representation for the on-particle $O_3$ - particulate BaP reaction was added to the model; and 4. process representation for in- and below-

cloud wet scavenging (including scavenging by snow) were added for gas-phase PAHs and benzene. Simulations using GEM-MACH-PAH were then carried out at high (2.5-km) spatial resolution in a small, but densely populated North American domain including southern Ontario, and most of the northeastern U.S. (Fig. 1) for summer and winter of 2009. We refer to this region as the "Pan Am" domain because it was created for high-resolution air quality modelling during the 2015 Pan Amer-

ican Games in Ontario (Joe et al., 2017). This domain contains approximately 109 million people, including about 38% of the Canadian population and 30% of the U.S. population. The model results were evaluated using measurements from a high-spatial-resolution campaign in Hamilton, Ontario (Anastasopoulos et al., 2012), as well as the binational Integrated Atmospheric Deposition Network (IADN), the Canadian National Air Pollution Surveillance network (NAPS), and the U.S. National

Air Toxics Trends Stations network (NATTS). We focus our model evaluation on spatial variations at high resolution, estimating the level of model sensitivity to uncertainties in the PAH emission factors, gas-particle partitioning, and wet deposition, which are all related to novel aspects of the GEM-MACH-PAH model.

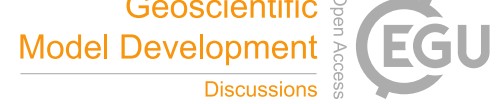

The following sections will further describe the GEM-MACH-PAH model (Section 2), the measurements used for evaluation (Section 3), the results of the model evaluation (Section 4), and conclusions (Section 5).

## 2   Model Description

In the present study, we have modified ECCC's high-resolution air quality forecasting model, GEM-MACH (hereafter called "GEM-MACH-PAH"), to include benzene and seven PAHs (in both gas and particle-phases) and have carried out 6 months of simulations in 2009 at the highest resolution (2.5-km grid cell size) in a North American domain yet reported for PAH simulations, to our knowledge. We have also tracked PAH wet deposition, and gas-particle partitioning, and have attempted to qualify model sensitivity to uncertainty in mobile emission factors, which has not been reported in other model studies.

### 2.1   GEM-MACH overview

GEM-MACH is an on-line, 3D chemical transport model, which is embedded in GEM, ECCC's operational numerical weather prediction model (Côté et al., 1998b, a; Moran et al., 2010). On-line models such as GEM-MACH improve air-quality chemical prediction performance by reducing interpolation errors between different model coordinate systems and removing the input/output time and disk storage required for the transfer of meteorological input files to their off-line CTM counterparts (e.g., Baklanov et al., 2014). The coupling to meteorology is a one-way process in this version, whereby chemistry does not influence the meteorology. More detailed description of the gas-, aqueous-, and particle-phase process representations of GEM-MACH, and an evaluation of its performance for common pollutants such as ozone, particulate matter (PM), and ammonia appears in Moran et al. (2013); Makar et al. (2015b, a); Gong et al. (2015), and Whaley et al. (2017). Here we will focus on the model changes made to include PAH species and processes.

GEM-MACH is used to provide ECCC's twice-daily, 48-hour operational public forecasts of criteria air pollutants (ozone, nitrogen oxides, PM), as well as the Air Quality Health Index [https://ec.gc.ca/cas-aqhi/]. To reduce the computational burden for forecasting, the PM size distribution is represented using a simplified sectional treatment consisting of two size bins, a fine-fraction bin for particles with Stokes diameter from 0 to 2.5 $\mu$m and a coarse-fraction bin for particles with Stokes diameter from 2.5 to 10 $\mu$m (Moran et al., 2010), with sub-binning used for those particle processes requiring a finer particulate size distribution. Here, we utilize the research version of GEM-MACH version 2, revision 2476, with this two-size-bin representation as our starting point for PAH modifications. The model grid used corresponds to a rotated latitude-longitude map projection with 2.5-km horizontal grid spacing and a hybrid vertical coordinate with 80-level vertical discretization spanning the atmosphere from the surface to 0.1 hPa.



## 2.2 Model modifications for benzene and PAH species

Our modifications to GEM-MACH include adding benzene and seven gas-phase and 14 particle-
phase (7 species × 2 size bins) PAHs to the species arrays, and adding the gas-particle partitioning
subroutine described in Galarneau et al. (2014), but with updated partitioning parameters (see Sec-
tion 2.2.1). Since PAHs have very small concentrations relative to criteria air contaminants, as in
Galarneau et al. (2014), we assume they do not have a significant effect on oxidant concentrations
($O_3$ and OH). Thus, the PAHs in GEM-MACH-PAH make use of the outcomes of the model's gas
and aqueous-phase chemistry in a diagnostic fashion for PAH oxidation. Processes in which the
PAHs participate directly include advection, vertical diffusion, plume rise of major point source
emissions, aerosol particle microphysics, in- and below-cloud scavenging, and dry and wet depo-
sition of both gas and particle phases. Some of these processes and/or their controlling parame-
ters were updated relative to Galarneau et al. (2014) and are described in the subsections below.
Like AURAMS-PAH, the total (gas+particle) PAH emissions were treated as gas-phase emissions
in GEM-MACH-PAH, since these quickly repartition between particles and gas phases following
emission. The non-PAH and PAH emissions are described further below.

### 2.2.1 Gas-particle partitioning

As PAHs are semi-volatile organic compounds that partition between the particulate and gaseous
phases of the atmosphere, their partitioning is a major determinant of their atmospheric fate (Bidleman,
1988). Despite decades of study (Junge, 1977; Yamasaki et al., 1982; Bidleman and Foreman, 1987;
Pankow, 1987; Smith and Harrison, 1996; Dachs and Eisenreich, 2000; Lohmann and Lammel, 2004;
Keyte et al., 2013), the mechanisms responsible for PAH partitioning and its spatiotemporal variabil-
ity are not well-understood. AURAMS-PAH included two parametrizations to calculate gas/particle
partitioning: Junge-Pankow, JP (Junge, 1977; Pankow, 1987) and Dachs-Eisenreich, DE (Dachs and Eisenreich,
2000), both of which, when applied for partitioning in AURAMS-PAH, assigned too much PAH
mass to the gas phase. The two schemes resulted in surprisingly similar gas-particle partition-
ing (Galarneau et al., 2014). We carried out post-processing and analysis on the AURAMS-PAH
model output from both schemes as well as the observations of gas and particle PAHs from the
Galarneau et al. (2014) study, in order to determine which scheme to proceed with in GEM-MACH-
PAH, and how it could be improved.

Measured PAH partitioning typically takes the linear form of:

$$logK_{p,k} = m_K logp_{L,k}^\circ + b_K, \tag{1}$$

where $K_{p,k}$ is the partitioning coefficient for each PAH species, k:

$$logK_{p,k} = log\left[\frac{C_p/C_{TSP}}{C_g}\right], \tag{2}$$





and $C_p$, $C_{TSP}$, and $C_g$ are the concentrations of the particulate PAH, the total suspended particles, and the gas-phase PAH, respectively. $p_{L,k}^{\circ}$ is the sub-cooled liquid vapour pressure of the k'th gas, and $m_k$ and $B_k$ are empirically derived coefficients. This linear relationship (where the $\log K_p$ of all PAH species in any given measurement sample fall on a line relative to their $\log p_L$) is com-
mon among homologous compound groups such as PAHs, polychlorinated biphenyls (PCBs), and polychlorinated dioxins and furans. However, the JP formulation only allows for $m_K$ = -1 (see Section B in the supplemental material for more detailed information and a derivation). Conversely, observation-based estimates show a wide variety of $|m_K|$ values that are usually less than 1 (e.g., Fig. B.1a in the Supplemental Material), and this could be the reason why the AURAMS-PAH JP
model results under-predicted the particulate fraction.

Therefore, we proceeded with the Dachs-Eisenreich formulation in GEM-MACH, but with improved parameters. The Dachs-Eisenreich (DE) partitioning formulation was adapted from work examining water-sediment partitioning (Dachs and Eisenreich, 2000). The DE expression for $K_p$ (Eq. (B.2.1) is related to the octanol-air and soot-air partitioning coefficients, the latter depending on the
soot-water ($K_{SW,k}$) and air-water partitioning coefficients. The soot-water partitioning coefficients are highly uncertain. Their values in the literature span two orders of magnitude for the same compound (Dachs and Eisenreich, 2000; Bucheli and Gustafsson, 2000; Jonker and Koelmans, 2002; Xu et al., 2012). $K_{SW,k}$ from Jonker and Koelmans (2002), was used in AURAMS-PAH. However, using the 2002 measurement data and their average $m_K$, we have determined new $K_{SW,k}$ values
based on ambient observations that improve the DE particulate fraction representation (see Section B.2 in the supplemental material for this process, and Table 1 for the original and new values). The purple boxes in Fig. 2 represent the results of the AURAMS-PAH partitioning module, making use of our new $K_{SW,k}$ values instead of the originals.

While the adjusted $K_{SW}$ values in Table 1 are significantly different from those in the original
model (based on Jonker and Koelmans (2002) as adjusted by the relative contributions of PM mass to the domain total in the inventory of Galarneau et al. (2007)), particularly for lower molecular-weight species, they fall within the range of values found in the literature [e.g., Dachs and Eisenreich, 2000; Bucheli and Gustafsson, 2000; Jonker and Koelmans, 2002; Xu et al., 2012].

### 2.2.2   Emissions

Chemical (non-PAH) emissions in GEM-MACH make use of data from the U.S. Environmental Protection Agency (EPA)'s 2011 National Emissions Inventory (NEI), and Canada's 2010 Air Pollutant Emission Inventory (APEI), these being the closest available inventory years to the year in which our simulations takes place (2009). PAH model emissions were created with the SMOKE emissions processing system (Sparse Matrix Operator Kernel Emissions, https://www.cmascenter.org/smoke/),
which utilized PAH-to-TOG (total organic gases) emission factors, that were originally compiled for AURAMS-PAH by Galarneau et al. (2007, 2014). Below we outline the further modifications





and updates that we made to this existing emissions database, in order to generate updated PAH emissions for modeling.

*PAH stationary emissions*

Most of the PAH emission factors (EFs) used for the 2002 AURAMS-PAH model were compiled from the U.S. EPA's Locating and Estimating Series (U.S. EPA, 1998), AP-42 document (US EPA, 1995), and the 1999 National Emissions Inventory (NEI99), (Galarneau et al., 2007). PAH EFs for stationary sources that were published between 1999 and the present are not substantially different from those already being used in SMOKE. For example, recent literature on agricultural burning

(e.g., Dhammapala et al., 2007; Hall et al., 2012) reported EFs that were close (within a factor of two) to those already in the inventory. Only emissions from iron and steel production were updated to those in Odabasi et al. (2009) for electric arc furnaces, as the values used in Galarneau et al. (2007) were derived from literature published before 1990, and were 1-2 orders of magnitude larger (hence likely represented outdated (or absent) pollution control equipment).

*PAH mobile emissions*

On-road mobile PAH emission factors in AURAMS-PAH were taken from NEI99 (Galarneau et al., 2007). The mobile emissions in this inventory may no longer be relevant as the values compiled were from an older (1990s) vehicle fleet. Therefore, we employed updated EFs for more current on-road mobile emissions in Canada and the U.S. for 2009 modelling. Also, some off-road emissions, such

as emissions from helicopter and marine (large ships) were not considered before, and were added to the inventory (from Chen et al. (2006) and Agrawal et al. (2008), respectively) in this study.

MOVES 2014, the latest version of the U.S. EPA's motor vehicle emissions simulator (www.epa.gov/moves) contains a more recent standard set of mobile EFs, separated into one set of factors for gasoline vehicles, based on one large, 2008 study of vehicles in the U.S. (Sandeep et al., 2008), and one set

of factors for diesel vehicles, based on another large study in the U.S. (Khalek et al., 2009). In order to investigate whether these U.S. values would be representative of conditions in Canada and whether only have those two fuel-type categories are adequate, when this would neglect studies that have reported different EFs for several different vehicle/fuel categories (e.g., cars, trucks, buses, motorcycles; light- or heavy-duty; gasoline or diesel), we compiled and researched PAH-to-TOG

emission factors for these classes of mobile sources from over 30 recent (1999 to present) publications, as well as from the U.S. EPA's SPECIATE v4.4 database (containing data from 1990 to 2012: https://www.epa.gov/air-emissions-modeling/speciate-version-45-through-40). Please refer to Section C in the supplemental material for this mobile emission factor analysis. In this analysis, we found that the MOVES2014 EFs provided the best results in the model, thus they were selected for

use in our simulations.

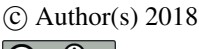



### 2.2.3 On-particle BaP-O$_3$ reaction

The only PAH oxidation reactions included in AURAMS-PAH were temperature-independent OH reactions with each gas-phase PAH species (Galarneau et al., 2014), which were also added to the GEM-MACH-PAH model. Temperature-dependent OH reaction rates were not pursued because Brubaker and Hites (1998) determined that only $k_{OH}$ for fluoranthene has a slight temperature dependence, but dependence was smaller than their error levels. Also, Friedman and Selin (2012) performed a phenanthrene sensitivity study with their model, and determined that including temperature dependence in $k_{OH}$ did not affect their mean non-urban mid-latitude concentrations.

The AURAMS-PAH model overestimated BaP concentrations compared to measurements (Galarneau et al., 2014). This could be due to two O$_3$-related factors: (1) Particulate BaP measurements are known to be affected by on-filter O$_3$ degradation, causing measured particulate BaP measurements to be biased low (Menichini, 2009); (2) Heterogeneous BaP degradation by O$_3$ in ambient air (Keyte et al., 2013) was not simulated in AURAMS-PAH, thereby biasing modelled concentrations high. We therefore added a particle-phase BaP-O$_3$ reaction in GEM-MACH-PAH to account for the latter atmospheric process as described next. For the former, on-filter O$_3$ reaction, we have attempted to correct the measurements as described in Section 3.

In GEM-MACH-PAH we used the Kwamena scheme (Kwamena et al., 2004) for the atmospheric on-particle BaP-O$_3$ reaction, as this scheme produced the best results in Friedman and Selin (2012)'s global model, and according to our sensitivity calculations, other schemes either overestimate (e.g., Pöschl et al., 2001) or underestimate (e.g., Kahan et al., 2006) the amount of BaP destroyed by this reaction. The Kwamena scheme was used because it produced BaP loss consistent with measurement studies (Ringuet et al., 2012; Jariyasopit et al., 2014; Liu et al., 2014). The reaction rate, $k$, is expressed as follows:

$$k = \frac{k_{max}K_{O_3}[O_3]}{1 + K_{O_3}[O_3]},$$
(3)

where, $k_{max} = 0.060 \pm 0.018$ s$^{-1}$ and $K_{O_3} = (2.8 \pm 1.4) \times 10^{-15}$ cm$^3$. The Kwamena scheme is expressed in the model as:

$$[BaP]_{reduced} = [BaP]_i e^{-k\delta t},$$
(4)

where $\delta$t is the model time step in seconds. This formulation does not depend on the particle size, but rather on the overall bulk particulate concentration, and the concentration of O$_3$.

### 2.2.4 Dry and wet removal of PAHs and benzene

Gas-phase dry deposition follows a multiple resistance approach and single-layer "big leaf" approach (Wesely, 1989; Zhang et al., 2002; Makar et al., 2018) with a temperature dependency for Henry's Law constants and for water solubility (Sander, 1999; Ma et al., 2010). Dry deposition of benzene





and PAHs is also output by the model; however, measurements of the deposition flux of these species
were unavailable during the study period.

    PAH particle-phase dry deposition is treated following (Zhang et al., 2001), resulting in size-
dependent particle deposition velocities.

    Gas-phase benzene and PAHs undergo cloud and rain scavenging via Henry's law. Henry's law
partition coefficients ($K_{AW}$) for the seven PAHs is a linear relationship with inverse temperature.
The mass of benzene and PAHs in the gas-phase (as opposed to the aqueous phase in cloud droplets
and raindrops) is derived from:

$$K_{AW,k} = \frac{m_{gas}/V_{air}}{m_{aq}/V_{h2o}} = m_{AW}/T + b_{AW}, \qquad (5)$$

and solving for $m_{gas}$:

$$m_{gas} = \frac{\frac{V_{air}}{V_{h2o}} K_{AW,k} m_i^{gas}}{\left(1 + \frac{V_{air}}{V_{h2o}} K_{AW,k}\right)}, \qquad (6)$$

where, $m_i^{gas}$ is the initial mass of the gas-phase PAH before the Henry's law partitioning, and the
remaining PAH mass is scavenged to the liquid rain or cloud phases ($m_{aq}$). Note that Okochi et al.
(2004) reported that assuming Henry's Law equilibrium for benzene underpredicts the extent of
wet-deposition. In the absence of a suitable alternative parameterization, we used Henry's Law par-
titioning and therefore obtained a conservative estimate of wet deposition for benzene. Note that
benzene wet deposition is not evaluated in this paper as no measurements are available.

    Where temperatures are $<0°C$ below-cloud, or $<-15°C$ in-cloud, scavenging of gas-phase ben-
zene and PAHs by snow and cloud-ice is done via surface adsorption following the formulation used
in Franz and Eisenreich (1998), which was also used by Wania et al. (1999); Lei and Wania (2004)
and Friedman and Selin (2012):

$W_g = K_{ia,k}(SA)\rho, \qquad (7)$

where $W_g$ is the gas scavenging ratio (equal to the concentration of PAH in snow over the concen-
tration of PAH in air - both in moles/m$^3$), $K_{ia,k}$ is the interfacial adsorption coefficient (equal to
the mass adsorbed per surface area of snow to the atmospheric vapor phase concentration - both in
ng/m$^3$), $SA$ is the specific surface area of the snow crystal, for which we use a constant 1 m$^2$/g based
on literature values for fresh snow precipitation, which are highly variable, and for which no clear
relationship with temperature or wind speed has been found (Hoff et al., 1998; Hanot and Dominé,
1999; Domine et al., 2007; Hachikubo et al., 2014), $\rho$ is the density of ice (0.917 g/cm$^3$), and $K_{ia,k}$
is calculated from the following (Franz and Eisenreich, 1998):

$$log(K_{ia,k}) = -1.2 log p_{L,k}^\circ - 5.82, \qquad (8)$$

Eq. (7) is used to determine the fraction of PAH mass in the gas and snow/cloud-ice phases.





Particle-phase PAHs are treated as passive tracers that undergo wet removal along with the modelled aerosol particles (Gong et al., 2006; Wang et al., 2010). The cloud and precipitation processes above are applied sequentially in the model using operator splitting, and the amount of PAH deposited from wet deposition is output by the model. Table D.1 in the supplemental material provides

all of the constants used in the model.

### 2.3   Model setup for two 3-month simulations

GEM-MACH-PAH, rev2488, was run from 8 May to 13 August 2009 and from 18 October 2009 to 5 January 2010, where the first week from each period is treated as a spin-up period (for chemical concentrations to stabilize and for the initial condition effects to be negligible: e.g., Samaali et al.,

2009), and were not used in our evaluation. The time periods were chosen to coincide with as many PAH concentration and deposition measurements as possible, while limiting simulation duration to reduce computational expenses.

The chemical initial and boundary conditions for the outer nest North American domain were taken from a one-year MOZART simulation for all pollutants (Emmons et al., 2010; Pendlebury et al.,

2017), except for benzene and PAHs. Initial and boundary conditions for PAHs and benzene were set to zero for the North American domain as its boundaries are generally away from PAH and benzene sources (e.g., over the ocean), and are also very distant from the Pan Am domain. The simulations for the nested Pan Am region were run using the chemical initial and boundary conditions from the 10-km North American model run.

The model simulation was carried out in sequence of 27-hour staggered simulations starting at 00 UTC, in order to reinitialize meteorology with the analysis at 10-km resolution. The first three hours in the 2.5-km domain were discarded as spin-up to reduce the dependency on the 10-km resolution meteorological initial conditions. Each 27-hour simulation used the chemical concentrations from the end of the previous simulation as initial conditions for the next 27 hours, and this sequence

continued until each 3-month period was complete.

GEM-MACH-PAH was run in the 2-size-bin mode to represent the PM size distribution, which means that particles fall in either fine mode ($PM_{2.5}$ - diameter 2.5 $\mu$m or less) or coarse mode ($PM_{10}$-$PM_{2.5}$ - diameter 2.5-10 $\mu$m).

### 3   Measurement Description

We compare the GEM-MACH-PAH predictions to all of the benzene and PAH measurements available in the Pan Am domain during the two time periods in 2009. These include a high-spatial-resolution urban measurement campaign in the Hamilton, Ontario region, as well as network monitoring stations from NAPS, NATTS, and IADN. Locations of PAH and benzene measurement stations are plotted in Fig. 3 and 4a. Note that all measurement stations were *not* equipped with oxidant



removal technology; therefore, all measured PAHs, especially benzo(a)pyrene (which has the high-
est particulate fraction, and is the most reactive with $O_3$), would have had losses due to reaction
with ozone on the filters (Menichini, 2009; Liu et al., 2014), and thus would be biased low com-
pared to concentrations in ambient air. Accordingly, we have applied an $O_3$ correction to the BaP
measurements in this study, as the literature suggests that the BaP sampling artifact is substantial,

with around 20-72% lost on average during sampling (Menichini, 2009; Liu et al., 2014). Note that
our correction follows the linear method recommended by Schauer et al. (2003), which is dependent
only on $O_3$ concentrations. However, other studies state that the $O_3$ degradation of BaP is more
complex, with additional dependencies on the resident atmospheric lifetime of BaP (Goriaux et al.,
2006), and relative humidity (Pitts Jr. et al., 1986; Umwelterhebungen and Gerätesichereit, 2002;

Menichini, 2009). However, those studies did not provide an alternative correction equation. There-
fore, in our results, we will present both the Schauer-corrected BaP measurements (for sites that had
$O_3$ monitors nearby), as well as the original reported BaP from the measurements, given the lack of
a better correction for the sampling artifact.

### 3.1 Hamilton measurement campaign

Ambient measurements of $PM_{2.5}$ and 16 PAH species were collected from a dense network of mea-
surement sites in Hamilton, Ontario during June/July 2009 and December 2009. These measure-
ments are described in Anastasopoulos et al. (2012), where they found a high level of intra-urban
variability for the PAHs; 3-4 times more variable than $PM_{2.5}$ concentrations.

There were 43 measurement sites operating during the summer period (see Fig 4a), and 46 sites

during the winter period. All measurements are from 2-week integrated time frames (24 June to
8 July and 2 to 16 December) taken with URG personal pesticide samplers, which collected gas
and particle-phase PAHs less than 2.5 $\mu$m in diameter in 40 m$^3$ of sample air. The $PM_{2.5}$ measure-
ments were made at the same sites using a three-stage Harvard Cascade Impactor. The particle-phase
PAHs (up to 2.5 $\mu$m in diameter) were collected on a Teflon filter, gas-phase PAHs were collected

in polyurethane foam (PUF), and total (gas + particle) PAHs concentrations were reported in ng/m$^3$
as determined by gas chromatography/mass selective detection. PM sample filter masses were deter-
mined by gravimetric analysis. (Anastasopoulos et al., 2012)

$O_3$ measurements that were used to correct the Hamilton BaP measurements came from three
monitoring sites in the Hamilton region ("Downtown", "Mountain", and "West") from the Ontario

Ministry of Environment and Climate Change (MOECC) website for historical air quality data
(http://www.airqualityontario.com/history/).

### 3.2 National Air Pollution Surveillance Program

NAPS is a Canadian program to provide accurate and long-term air quality data of a uniform stan-
dard across the country. NAPS is managed under a cooperative agreement between ECCC and the



provinces, territories, and some municipal governments. There are currently 286 NAPS measure-
ment sites in 203 communities located in every province and territory [www.ec.gc.ca/rnspa-naps/].

Under this program, PAH samples were collected over 24 hours, beginning and ending at midnight
(local), typically every 6 days, with a sample volume range of 600-800 m$^3$ (Environment Canada,
2013). Benzene samples were collected in 6-L stainless steel canisters over 24 hours, starting at
midnight, every 3 days (Galarneau et al., 2016).

Within the Pan Am domain, total PAHs (gas+particle-phases combined) and benzene were mea-
sured at eight NAPS sites (listed in Table A.1 in the supplemental material; Fig. 3), and their 2009
data were downloaded from the following url: http://maps-cartes.ec.gc.ca/rnspa-naps/data.aspx.

For BaP measurement corrections, the NAPS network also measures hourly O$_3$ at four of these
eight PAH/benzene sites (Windsor, Hamilton, Simcoe, and Egbert). Two of the missing sites (Toronto
and Etobicoke) had nearby O$_3$ measurements from MOECC, but the last two (Burnt Island, and
Point Petre), which are rural sites, had no O$_3$ measurements nearby. Therefore, BaP could only be
corrected at six of the eight NAPS sites in the Pan Am domain.

### 3.3 National Air Toxics Trends Stations network

NATTS is an a U.S. program to monitor toxic air pollutants in accordance with the U.S. Government
Performance Results Act, which requires the U.S. EPA to reduce the risk of cancer and other serious
health effects associated with hazardous air pollutants (HAPS) by achieving a 75% reduction in air
toxics emissions chemicals, based on 1993 levels (U.S. EPA, 2009). Regulated under the Clean Air
Act are 188 HAPS species including benzene and the seven PAHs in this study.

Every six days, 24-hour ambient air samples are collected starting at midnight LT. Analysis of the
samples is done by high resolution gas chromatography/mass spectrometry (GCMS) Selective Ion
Monitoring (SIM) mode to get total (gas + particle) PAH concentrations, and benzene concentrations
(Eastern Research Group, Inc., 2009).

There are 115 NATTS sites within the model domain (Table A.1 in the supplemental material,
Fig. 3), but only 21 sites measured PAHs, while 113 sites measured benzene. Those data were down-
loaded from the following url: www.epa.gov/ttnamti1/toxdat.html#data. Measurement methods in
NATTS are very similar to those of NAPS.

Since O$_3$ was not measured at the NATTS sites, NATTS BaP was corrected with the nearest O$_3$
monitor data found at the U.S. EPA and CASTNET websites: www.epa.gov/outdoor-air-quality-data/download-daily-data
and https://java.epa.gov/castnet/reportPage.do, respectively.

### 3.4 Integrated Atmospheric Deposition Network

IADN was mandated by the 1987 Canada-U.S. Water Quality Agreement, and was initiated in 1990
to measure atmospheric concentrations of persistent toxic pollutants in the Great Lakes basin. There
are nine IADN sites total within our Pan Am model domain, and they are listed in Table A.1 of the





supplemental material (see also Fig. 3). Six of the nine IADN sites report gas- and particle-phase PAH atmospheric concentrations separately (labelled "PAHs" in Table A.1), and a different set of six sites report wet deposition of PAHs (labelled "PAH wet dep" in Table A.1) using sampled precipitation concentrations (Blanchard et al., 2005). Thus, these data can be used to evaluate the model's gas-particle partitioning and deposition, respectively. Benzene was not measured by IADN, nor was

$C_{TSP}$ or $PM_{10}$ in 2009, with the unfortunate result that $K_p$ can not be calculated directly from the IADN measurements. $O_3$ was also not measured by IADN, necessitating the use of the nearest $O_3$ monitors in order to carry out the BaP oxidation correction. This latter step was possible only for observation stations at Cleveland and Chicago. The other four IADN air sites were rural/background locations, and did not have any $O_3$ measurements nearby.

PAHs were collected by high-volume sampler for periods of 24 hours beginning at 08:00 Eastern Standard Time, every 12 days. At Canadian IADN sites, glass fiber filters and PUF sorbent collected the particulate and gaseous fractions, whereas the U.S. stations collected PAHs with quartz fiber filters and XAD resin (Blanchard et al., 2005). Sample volume for the U.S. method is about 800 m$^3$, but is 400 m$^3$ for the Canadian method to minimize breakthrough of volatile species during warm

summer months (Blanchard et al., 2005).

    Wet deposition of PAHs are measured with MIC-B precipitation collectors. The U.S stations used XAD-2 resin column cartridges for accumulating the organics on a 28-day cumulative basis, while the Canadian stations use a dichloromethane solvent extraction system, also on a 28-day cumulative basis. Both countries collect samples on a monthly basis. Note that one of the six wet deposition

sites, St Clair, Ontario (STC), only had valid measurements during February 2009, which was not a time period simulated here. Therefore, only five IADN sites appear in our wet deposition analysis in Section 4.4 below.

    Note that Point Petre and Burnt Island are NAPS stations co-located with IADN. IADN data were downloaded from the following url: http://open.canada.ca/data/en/dataset/531d6054-4179-4883-8022-1175cdfb6911.

## 425  4   Model Evaluation

In this section we evaluate GEM-MACH-PAH's performance for benzene and PAH surface concentrations, their spatial variation, gas-particle partitioning, and wet deposition. We also assess the sensitivity of the model output to PAH emission factors for mobile sources.

### 4.1   PAH concentrations in the Hamilton region

GEM-MACH-PAH output for gas + fine-PM PAH were compared to measurements of same from the 2009 Hamilton campaign (Anastasopoulos et al., 2012). Fig 4a shows a map of measured and modelled fluoranthene concentrations (14-day average) in the summer time period, as well as their differences and ratios. Here we see that GEM-MACH-PAH has captured intra-city variability, and





that the differences between observations and simulated values are, at a maximum, $2.8\times$ too high.

The model is biased low in the upwind/background areas of the city, and a high in the eastern areas of the city (Fig. 4a), and this pattern is seen across all seven PAHs. The spatial pattern in the PAH bias is less apparent when PAH/PM$_{2.5}$ ratios are plotted (in ng/$\mu$g – shown in Fig. E.1 in the supplemental material) – removing the dependency on modelling PM correctly (since fractions of the PAH are particulate). Therefore, the spatial pattern in the PAH bias is mainly due to the pattern in the model

PM bias, which is shown in Fig. E.2 in the supplemental material.

When the spatial variability is represented by the standard deviation over the mean ($\sigma$/mean), the model achieves very similar spatial variability to the measurements (Fig. 4b). The scatter plot of model vs measurements for summertime fluoranthene concentrations (Fig. 5a, FLRT selected as a good example) has correlation coefficient $R^2$ of 0.57, and the slope of the best-fit line is very close

to 1. The other PAH species had similar results, where, except for FLRT, the slopes and $R^2$ values were better in the winter than in the summer.

The model bias (given as a model/measurement ratio) for all PAHs is shown as box and whiskers in Fig. 5b. Here we see that wintertime biases are smaller than those in the summertime for all PAHs except for ANTH. The four lightest PAHs (left side of Fig. 5b) have model/measurement ratios near

1 (except summertime PHEN), but the three heaviest PAHs, are biased high (except for wintertime CHRY). We will see this same pattern for the model bias (small for lighter PAHs, high for BaA and BaP) in the next sections as well.

BaA and BaP are the most reactive of the heavier species, thus the lack of O$_3$ correction to the BaA measurements may be partially responsible for the model-measurement differences. However,

mean O$_3$ for the three measurement stations in the Hamilton region was only 20-26 ppbv/day, thus the O$_3$-corrected BaP was approximately 20% greater than the reported BaP concentrations. The median BaP bias was brought down to 6.2 from 7.6 in the summer, and 5.5 from 6.3 in the winter - these are shown as the purple boxes in Fig. 5b. Additional reactions with BaA and BaP, such as with NO$_3$, are noted in the literature (Keyte et al., 2013; Mu et al., 2017), but were not included in

GEM-MACH-PAH at this time, given larger uncertainties in those reactions. However, our model biases appear to indicate that those missing reactions may need to be considered for further model improvement.

In order to remove the impact of the model's PM predictions on the PAH comparison, we also plotted the PAH/PM$_{2.5}$ model-over-measurement ratios (shown in Fig. E.3). There we see all of the

ratios reduced – which improves results for the heavier PAHs, but increases the low bias for the lighter PAHs. The reason the bias decreases for all seven PAHs is that the model PM$_{2.5}$ is overestimated by a factor of 2 in the summertime, and a factor of 1.4 in the wintertime (average across all sites in the Hamilton region).



### 4.2 PAH and benzene concentrations from the NAPS, NATTS, and IADN networks

Modelled 24-hour-average total (gas+particle) PAH and benzene concentrations can be evaluated with every-$6^{th}$-day measurements from the NATTS, NAPS, and IADN surface measurement networks, which sample much of the model domain well (Fig. 3). As with the Hamilton evaluation, the model has very good agreement for seasonal averages at the monitoring network sites for benzene, phenanthrene, anthracene, fluoranthene, and pyrene, which all have model/measurement ratios (red

and blue boxes) close to 1, and their concentrations (green and orange boxes) overlapping in Fig. 6a. The 24-hour average model (daily) and measurements (every $6^{th}$ day) have been averaged over each of the 3-month time periods. BaA and CHRY are within a factor of 5 of the measurements in the summertime, but worse in the wintertime. BaP is overestimated by the model in both summer and winter by about a factor of 10, although the measurement-corrected BaP has a slightly reduced bias.

We have not shown the $O_3$-corrected BaP measurements in the plots because the changes are small, similar to the Hamilton plot (Fig. 5a).

   When the model biases are examined more closely we find a few patterns to determine the cause(s). The following list summarizes some observations from our evaluation of each model-measurement pair (24-hour averages, not seasonal averages):

– **By site - overestimations**: all PAHs are significantly overestimated at the Kennedy Township, Pennsylvania (NW of Pittsburgh) NATTS site (Fig 6b). There appears to be a major emissions point source near that station that is emitting too much PAH in our model compared to reality. There are in fact hundreds of point sources in the emissions inventory that are within 20 km of Kennedy Township, but one in particular emits a relatively large amount of VOCs, and is

associated with the "Secondary Metal Production; Aluminum; Raw Material Charging" source category, which has very large PAH-to-TOG EFs in Galarneau et al. (2007) because aluminum smelter emissions are largely particulate, so expressing EFs as a large fraction of TOG was somewhat artificial. However, our results indicate that the PAH EFs for that PAH speciation profile (1036b) should be reduced substantially compared to Galarneau et al. (2007). In order

to ensure that this facility did not begin operation *after* 2009 - which is a risk when using a 2011 inventory to model the year 2009 and would result in a large overestimation as well, we have further verified that the facility existed and was emitting similar VOC amounts in the NEI2008 inventory as well.

   PHEN (Fig. 7c) and ANTH (not shown) are also greatly overestimated in New York City,

however, none of the other PAHs are biased particularly high there. However, we note that the measurements for New York City appear erroneously low, as the reported PHEN concentrations there are around the same magnitude as those in Underhill, VT (Fig. 7c), which is a background site, near a national park.



Most PAHs are also overestimated at the Gary, Indiana site (Fig 6b)), which may also have a nearby major point emissions source that is too high compared to reality. The heavier PAHs (BaA, CHRY, and BaP) are also overestimated at the Toronto Gage Institute NAPS site, but are only slightly higher there than the average model/measurement ratio for those species (not shown).

– **By site - underestimates**: all PAHs are markedly underestimated at the Liberty, Pennsylvania site (e.g., Figs. 6b, and 7c), implying that there may be industry emissions of PAHs here that are missing, mis-allocated, or misplaced in the NEI2011 inventory, or an improper PAH speciation profile applied. Similarly, PAHs are underestimated in Buffalo, New York and at Franklin Furnace, Ohio. As these are not large cities, there may be industrial emissions that are not reported in the NEI2011 emissions inventory (or are reported at too low levels) - perhaps because those facilities shut down in 2010 (or installed emission control technology), which would mean the problem simply lies in using a 2011 inventory to model 2009.

However, when we further investigated stacks near Buffalo, NY, we found that the facility with the largest CO and VOC emissions had zero PAH emissions. This facility is associated with the generic process of "Primary metal production; By-product Coke Manufacturing", which did not have an associated PAH-to-TOG profile in Galarneau et al. (2007), because the source category codes that follow it (such as flushing liquor circulation tank, excess-ammonia liquor tank, tar dehydrator, tar interceding sump, tar storage, etc) are not expected to emit PAHs to air. However, our results imply that the PAH speciation profile for "By Product Coke Oven Stack Gas" (0011b) would have been more appropriate for this facility and its use might eliminate the model bias near Buffalo in future studies.

– **By month**: All PAH species have lower mod/meas ratios in the summer than in the winter (shown in Fig. 6a by season for all PAHs and in Fig. 6c for FLRT by month) – implying that modelled hydroxyl radical (OH) and/or PM biases (which have strong seasonal cycles) are impacting modelled PAHs. For example, if model OH is too high in the summer, or too low in the winter, this would cause the U-shaped pattern that we see when plotting model/measurement ratio vs. month (Fig. 6c) and it would be particularly pronounced for the lighter, gas-phase PAHs, which it is. Another possibility is seasonal bias in the representation of atmospheric vertical stability: if the modeled stability is too low in the summer and too high in the winter, then winter emissions will tend to be trapped in inversions more than observed, and summer emissions will be diluted by excessive vertical mixing. However, evidence in Makar et al. (2010) and Stroud et al. (2012) suggest that model stability is too high (not too low) for the summer time period in those studies.

– **By season**: For the four lightest PAHs, the model/measurement ratios are <1 in the summer, and >1 in the winter (Fig. 6a). As mentioned above, this is likely due to modelled OH being





too high in the summer and too low in the winter. BaA and CHRY follow a similar seasonal difference but do not straddle the ratio=1 mark.

BaP, on the other hand has a model/measurement ratio that is slightly higher in the summer than in the winter (Fig. 6a). For BaP, the OH bias could be offset by an opposite $O_3$ bias in the model. Indeed, it has been shown (Makar et al., 2010; Stroud et al., 2012) that the processes

in GEM-MACH cause urban, surface $O_3$ to be too low in the summertime, due to insufficient vertical mixing and excessive titration from $NO_x$, and surface PM tends to be too high in the wintertime due to overestimation of wintertime atmospheric stability (e.g., lack of an urban heat island parameterization in the driving meteorology). These factors, together with the BaP measurement bias due to on-filter reaction with $O_3$, may explain the high model BaP bias.

Thus, the generally high bias of modelled BaP may to be due to additive OH, $O_3$ and PM model biases (plus the missing $O_3$ denuder technology in the measurements), impacting BaP more than the other species because BaP has the highest $O_3$ reactivity, and the highest particulate fraction of the seven PAHs examined here.

When the five measurement sites mentioned in "By Site", above (Kennedy Township, PA; Gary,

IN; Liberty PA; Buffalo, NY; Franklin Furnace, OH) are removed from the NATTS analysis (because errors in their nearby emissions were identified), model-measurement correlation (R) and slopes improve. For example, the model vs. measurement best-fit-line slope for PHEN doubles from 0.3 to 0.6 when those sites are removed, and its R increases from 0.16 to 0.32. The slopes and R values of the four heaviest PAHs all move from *negative* to positive. PYR has the largest improvement,

going from slope=-0.049 and R=-0.028 to slope=0.26 and R=0.35. To the extent that the model prediction errors at the other sites may reflect emissions inaccuracy, having an accurate major point emission inventory for the time period modelled, along with proper PAH speciation profiles are extremely important requirements for modelling PAHs well at high resolution. The cases with large discrepancies mentioned above highlight the need to be as specific as possible when assigning source

category codes to facility processes (which is difficult given that there are tens of thousands of point sources in the inventories).

That said, when using a paired t-test on all data to examine whether the summertime and wintertime modelled averages are the same as the measured averages, we found that the model was indistinguishable from the measurements for all PAH species (t<1 and p>0.05), *except* for win-

tertime BaP (which has t>1, and p<0.05, however, even with the $O_3$-corrected measurements). At finer time scales (e.g., *daily* model-measurement pairs) only modelled ANTH was statistically indistinguishable from measurements. Therefore, GEM-MACH-PAH can accurately model benzene and PAHs seasonally, but not daily.





### 4.2.1 Sensitivity of model to mobile emission factors

As discussed in the previous section, ensuring the accuracy of major point source emissions is important for model-measurement agreement near industrial locations. However, those major point source emissions tend to be located far from large population centres where human exposure is concentrated. In our inventory, mobile emissions make up 44%, 45%, 19%, 32%, 14%, 21%, and 30% of total PAH emissions, for PHEN, ANTH, FLRT, PYR, BaA, CHRY, and BaP, respectively, in our con-

tinental model domain, and studies have shown that the bulk of emissions within population centers is likely to originate from on-road mobile sector emissions (Dunbar et al., 2001; Pachón et al., 2013; Kuoppamäki et al., 2014; Miao et al., 2015). Thus, in order to accurately model ambient PAHs in urban centres, the uncertainty in emission factors from on-road vehicles may play a more significant role than major point sources.

We have thus carried out 2-week sensitivity simulations (9-24 May and 18 Oct-2 Nov 2009) wherein the mobile emissions of PAHs were scaled by factors of 0.5 and 1.5. This is approximately equivalent to the 25th and 75th percentiles in the range of emission factors found in the recent literature.

In Fig. 7, we show the surface PHEN time series from the measurements, base model run, and 0.5

and 1.5 scaled model runs at the IADN, NAPS, and NATTS sites. It is clear that – while a relatively small PAH source overall – changes to mobile emissions makes a large change in ambient PAH concentrations at certain urban locations, such as Philadephia, PA, New York, NY, and Burlington, Etobicoke, and Windsor, ON.

On average, there is about a 20-30% increase in PAH concentrations when mobile emissions

are increased by 50%, and a 5-10% decrease in PAH concentrations when mobile emissions are decreased by 50% (Fig. 8, PHEN and BaP shown as examples) – with a larger sensitivity in the summer than the winter, and slightly larger sensitivity at NATTS (U.S.) sites than at NAPS (Canada) sites. The predicted ambient concentrations generally follow the increase or decrease in on-road mobile emissions monotonically.

### 4.3 Gas-particle partitioning of PAHs


The IADN network also allows us to assess model predictions of gas-particle partitioning of PAHs at six sites (24-hour averages, every six days). Fig. 9 shows a time series of pyrene particulate fraction ($\phi_k$). Both model and measurements show higher $\phi_k$ in the wintertime, when there are higher PM concentrations for PAH adsorption, and lower temperatures. Generally, the model seems

to underestimate $\phi_k$ at background sites (e.g., Burnt Island), and overestimate $\phi_k$ at urban sites (e.g, Chicago), and this is true for all PAH species. Thus, in Fig. 10, which shows the results for all PAHs at all sites, the model (green) has a larger range of $\phi_k$ than the measurements (orange). This is caused by the model over- and underestimating PM concentrations at urban and rural sites,



respectively. For example, wind-blown dust is not included in the model; however, it is known to
be a potentially significant contributor to total PM in rural areas. Also, due to an underestimate of
vertical mixing in the model, PM tends to be biased high in urban areas, near emissions, due to a
lack of a parameterization for urban heat islands (Stroud et al., 2012).

Comparing Fig. 10 (blue boxes) to Fig. 2b (green boxes), we see significant improvement over
the original AURAMS-PAH partitioning due to the improved $K_{SW}$ parameters described in Section
615  2.2.1.

Generally, the gas-particle partitioning scheme in the model results in model/measurement ratios
well within an order of magnitude, given by the gray lines in Fig. 10. $\phi_k$ for BaA and CHRY are
still underestimated, but this may be related to modelled PM errors as noted earlier. We note that
the addition of $C_{TSP}$ or even PM$_{10}$ measurements at IADN sites (which existed in 2002, but not in
2009) would allow for the calculation of measured partitioning coefficients ($\log K_p$, Eq. (2)), which
could be used to validate the modelled $\log K_p$ in future work. Since $K_p$ takes total suspended particle
into account, it removes the dependency on modelled PM, thus would increase confidence that the
modelled partitioning is working properly, despite model errors in PM.

The fact that the GEM-MACH-PAH model partitioning of BaA and CHRY (and BaP to a lesser
extent) puts too much concentration in the gas phase, may help explain why these species in particu-
lar are overestimated in the model. While in the gas phase, these species are less likely to be removed
from the atmosphere, so their concentrations would erroneously build up in the model.

### 4.4 Wet deposition of PAHs

When compared to the IADN one-month wet deposition measurements, the model generally over-
estimates wet deposition for all PAHs, as is shown in Table 2, and Fig. 11. In Fig. 11, the blue
lines shows the ideal 1:1 model:measurement ratio, and most of the data lie well above these lines.
By site (Fig. 11a), the modelled wet deposition was slightly better at urban locations (Toronto and
Cleveland) than suburban and background sites (Burlington, Sturgeon Point, and Point Petre). By
month (Fig. 11b), the wet deposition from the model is best represented in June and July, whereas,
wet deposition is greatly overestimated in the winter, implying that the current snow adsorption
parameterization may be too effective at removing PAHs in the model.

The IADN measurements, which report the concentration of PAHs in the collected rainwater (in
pg/L), were converted to pgPAH/m$^2$ in order to compare to the wet deposition output of the model.
However, this conversion assumes that the volume of rainwater reported by IADN was the *total* rain
fall in the container's cross-sectional area, and in fact, it is not. The IADN wet deposition collectors
are actually known to *not* sample all of the rainfall because the samplers aren't in the correct configu-
ration to get an accurate rainfall measurement (Dryfhout-Clark, personal communication). When we
compared the actual rainfall amounts (from separate meteorological rain gauge data) to IADN rain
volumes at the Point Petre location in January 2009, we found that only 68% of the total rainfall was



captured by the wet deposition sampler. Therefore, if that correction factor were applied to all IADN wet deposition measurements, they would increase by a factor of approximately 1.5, which would improve our comparison, but not eliminate the total bias. If IADN sites added separate, accurate rain gages, then we could apply a "rainfall correction" to the IADN wet deposition measurements in a thorough, consistent way in future work.

Aside from the measurement bias, the modelled PAH wet deposition bias will also be dependent on the model's overall ability to predict accurate rainfall. We compared the modelled daily accumulated precipitation to the precipitation measured with the accurate gages at Burnt Island and Point Petre, and found that, while the model's median precipitation bias was only about 0.2 mm, there was a large standard deviation, and there were some incidences where the model greatly over-predicted
high rain events. Those incidences would result in greater modelled wet deposition of PAHs than was measured, and because we sum over a month, there is a significant likelihood of an overprediction occurring in that long time frame. Indeed, the median ratios in Table 2, which are less sensitive to high outliers than the mean is, are substantially lower than the mean for most species.

Therefore, the model bias in wet deposition would appear to be caused by three additive factors:
(1) measurements themselves having a negative bias relative to reality, due to insufficient capture of the net fluxes of precipitation, (2) modelled precipitation being biased high, and (3) a positive model bias in atmospheric PAH concentrations (which was highest for BaA and BaP in particular).

The reverse reasoning can be applied, whereby we can see if high atmospheric concentrations of BaA and BaP were caused by too little wet deposition. In this case, since both of these species have
correspondingly high wet deposition in the model (Fig. 11), it would appear that underestimation of wet deposition is probably not one of the causes.

Fig. 12 shows results from a sample month (June 2009) for pyrene (PYR). The spatial distribution of wet deposition was not captured, with the model predicting lower PYR deposition in Toronto and Sturgeon Point than the measurements, higher at Point Petre, and about equal at Burlington. This
spatial pattern is *not* the same for all PAHs, and even differs by month for the same PAH (e.g., PYR deposition in the next month, July, is low at Point Petre, high at Toronto, and highest in Burlington). Given that wet deposition of PAHs relies on getting many model factors correct (meteorology, scavenging parameters, atmospheric concentrations, etc), it not surprising that the model error for PAH wet deposition is large, but it is at least promising to see that there are no particular sites where the
model is consistently too high or too low, rather the errors in spatial distribution are haphazard and may be due to propagation of error, rather than any major error with the PAH scavenging scheme itself.





## 5 Conclusions

Through this work, a high resolution chemical transport model for North American air toxics was
created that allows us to see variations within a densely populated area. GEM-MACH-PAH was developed and run at 2.5-km resolution for air quality forecasting and for simulating the impacts of emissions scenarios. Relative to AURAMS-PAH, on-road mobile emissions, gas-particle partitioning, and scavenging were all improved in this study. Mobile PAH emission factors from different sources were evaluated and the MOVES 2014 factors achieved the best model results compared to
those in the recent literature and in the SPECIATE database. Parameters used in the gas-particle partitioning scheme (particularly $K_{SW,k}$) were improved based on the observed relationship between $\log K_p$ and $\log p$, resulting in much better agreement between model and observations than was achieved with AURAMS-PAH. This is an important improvement because the particle/gas partitioning determines deposition and inhalation - both pathways of exposure in humans and ecosys-
tems. Finally, we added snow scavenging, which was not a process included in AURAMS-PAH, and updated wet scavenging parameters.

Overall, GEM-MACH-PAH simulates benzene and six semi-volatile PAHs (PHEN, ANTH, FLRT, PYR, BaA, CHRY) at seasonal time-scales with concentrations statistically indistinguishable from observations, at 2.5-km resolution. For the seventh PAH species, BaP; its summertime average is
simulated to a similar level of accuracy. However, it appears the model's OH, $O_3$, and PM biases were additive, resulting in a wintertime average that is biased significantly high for BaP. Lack of removal of BaP via wet deposition was ruled out as a cause, but the lack of an $O_3$ denuder system in the measurements contributes a small amount to the model-measurement differences as well. When we corrected BaP measurements using the Schauer et al. (2003) $O_3$ relationship, we found reductions
of about 20% in the model/measurement ratios, improving the model performance.

Our results have shown that the major point source emissions play a large role in producing accurate model results near industrial facilities, but also that the uncertainty associated with on-road mobile emission factors plays a large role in the accuracy of simulations near and within cities. In fact, we have determined from our sensitivity test that the GEM-MACH-PAH model has a linear re-
sponse to a 50% variation in mobile emission factors, simulating concentrations that vary up to 30%. The spatial variability at high resolution is modelled to within 50% of Hamilton, Ontario measurements, although the model places higher concentrations in polluted areas, and lower concentrations in background areas than the measurements suggest, which is correlated to the spatial distribution of the model's PM bias. With this information, we can use the high-resolution GEM-MACH-PAH
model for studying vehicle emissions scenarios in order to determine intra- and inter-city variations due to motor vehicles, with an understanding of the range of uncertainty that such a study would have.

Additional improvements to PAH modelling efforts could be achieved with general model improvements to its treatment of particulate matter (e.g., better parameterizations for wind-blown dust





in rural areas, and better parameterizations for urban heat islands in urban areas). Also, additional
reactions with particulate BaA and BaP in the model (e.g., with NO$_3$) may reduce their bias further.
It would also be beneficial to any future model/measurement studies, if the PAH measurement networks utilized ozone denuder technology so that particle-phase PAHs are not underestimated in the
reported observations, as well as improved and consistent rain collection at wet deposition measure-
ment sites. Partitioning could be better assessed if sites that measure PAH gas and particle phases
separately (like IADN in this study) also measured $C_{TSP}$ or PM$_{10}$.

## 6 Data and code availability

Data availability: Please refer to Section 3 for the websites where the observations can be freely
downloaded.

Model code availability: GEM-MACH - Atmospheric chemistry library for the GEM numerical atmospheric model Copyright (C) 2007-2013 - Air Quality Research Division and National Prediction
Operations division, Environment and Climate Change Canada. This library is free software which
can be redistributed and/or modified under the terms of the GNU Lesser General Public License as
published by the Free Software Foundation; either version 2.1 of the License, or any later version.
The CHEM code can be downloaded from this Zenodo site: https://zenodo.org/record/1162252#.Wm9DtK1lJZQ,
DOI:10.5281/zenodo.1162252.

*Acknowledgements.* The authors gratefully acknowledge funding from ECCC's Climate Change and Air Pollution program (CCAP, formerly called the Clean Air Regulatory Agenda). Thank you to Angelos Anastasopolos and Amanda Wheeler for the Hamilton measurement data, to Helena Dryfhout-Clark for consultation on IADN
data products, and Armaan Ladak for his help with the measurement data. Figures were made using the R-based
OpenAir package (Carslaw and Ropkins, 2012; Carslaw, 2015).





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

**Table 1.** Original (from Jonker and Koelmans (2002)) and adjusted (based on AURAMS-PAH model-measurement analysis for North America) $K_{SW}$ values.

|              | PHEN   | ANTH   | FLRT   | PYR    | BaA    | CHRY   | BaP    |
|--------------|--------|--------|--------|--------|--------|--------|--------|
| Original Ksw | 4.34E5 | 1.55E6 | 2.24E6 | 1.70E6 | 3.74E7 | 2.82E7 | 9.59E7 |
| Adjusted Ksw | 3.32E7 | 4.21E7 | 8.24E7 | 9.84E7 | 1.75E8 | 3.23E7 | 1.10E8 |

US EPA: AP 42, Fifth Edition, Compilation of Air Pollutant Emission Factors, Volume 1: Stationary Point and Area Sources, Report, Office of Air Quality Planning and Standards, Office of Air and Radiation, Research
Triangle Park, NC, 27711, USA, 1995.

U.S. EPA: Locating and estimating air emissions from sources of polycyclic organic matter, Report epa-454/r-98-014, U.S. EPA, OAQPS, Washington, DC, 1998.

U.S. EPA: National Air Toxics Trends Stations quality assurance annual report calendar year 2009 - Final, Report, Office of Air Quality Planning and Standards, Research Triangle Park, NC 27711, USA, 2009.

Wang, X., Zhang, L., and Moran, M.: Uncertainty assessment of current size-resolved parameterizations for below-cloud particle scavenging by rain, Atmos. Chem. Phys., 10, 5685–5705, doi:10.5194/acp-10-5685-2010, 2010.

Wania, R., Mackay, D., and Hoff, J. T.: The importance of snow scavenging of polychlorinated biphenyls and polycyclic aromatic hydrocarbon vapors, Environ. Sci. Tech., 33, 195–197, doi:10.1021/es980806n, 1999.

Wesely, M. L.: Parameterization of surface resistances to gaseous dry deposition in regional-scale numerical models, Atmos. Environ., 23, 1293–1304, doi:10.1016/0004-6981(89)90153-4, 1989.

Whaley, C., Makar, P. A., Shephard, M. W., Zhang, L., Zhang, J., Zheng, Q., Akingunola, A., Wentworth, G. R., Murphy, J. G., Kharol, S. K., and Cady-Pereira, K. E.: Contributions of natural and anthropogenic sources to ambient ammonia in the Athabasca Oil Sands and north-western Canada, Atmos. Chem. Phys. Discussions,
2017, 1–38, doi:10.5194/acp-2017-627, 2017.

Xu, H.-Y., Zou, J.-W., Min, J.-Q., and Wang, W.: A quantitative structure–property relationship analysis of soot–water partition coefficients for persistent organic pollutants, Ecotox. Environ. Safety, 80, 1–5, doi:10.1016/j.ecoenv.2012.02.002, 2012.

Yamasaki, H., Kuwata, K., and Miyamoto, H.: Effects of ambient temperature on aspects of airborne polycyclic
aromatic hydrocarbons, Environ. Sci. Technol., 16, 189–194, doi:10.1021/es00098a003, 1982.

Zhang, J., Wang, P., Li, J., Mendola, P., Sherman, S., and Ying, Q.: Estimating population exposure to ambient polycyclic aromatic hydrocarbon in the United States - Part II: Source apportionment and cancer risk assessment, Environment International, 97, 163–170, doi:10.1016/j.envint.2016.08.024, 2016.

Zhang, J., Li, J., Wang, P., Chen, G., Mendola, P., Sherman, S., and Ying, Q.: Estimating population exposure to
ambient polycyclic aromatic hydrocarbon in the United States - Part I: Model development and evaluation, Environment International, 99, 263–274, doi:10.1016/j.envint.2016.12.002, 2017.

Zhang, L., Gong, S., Padro, J., and Barrie, L.: A size-segregated particle dry deposition 270 scheme for an atmospheric aerosol module, Atmos. Environ., 35, 549–560, 2001.

Zhang, L., Moran, M., Makar, P., Brook, J., and Gong, S.: Gaseous Dry Deposition in AURAMS A Unified Re-
gional Air-quality Modelling System, Atmos. Environ., 36, 537–560, doi:10.1016/S1352-2310(01)00447-2, 2002.





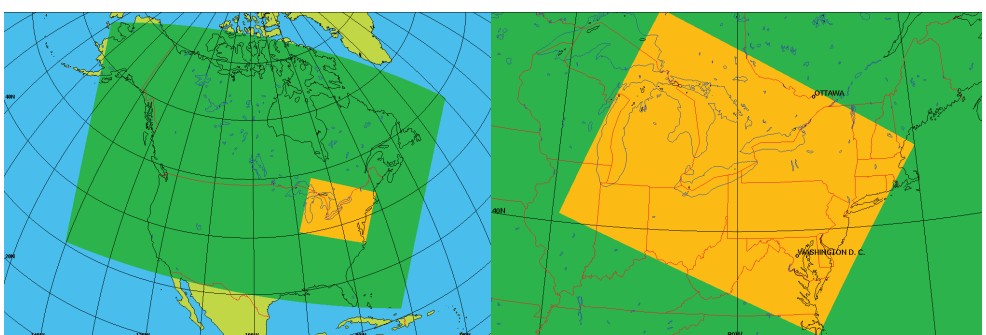

**Figure 1.** North American model domain with 10-km horizontal grid spacing (green), and the nested "Pan Am model" domain with 2.5-km horizontal grid spacing (orange).

**Table 2.** Mean and median GEM-MACH-PAH model/measurement ratios for PAH wet deposition

|  | PHEN | ANTH | FLRT | PYR | BaA | CHRY | BaP |
|---|---|---|---|---|---|---|---|
| mean ratio | 17.5 | 47.4 | 11.5 | 7.4 | 22.2 | 6.1 | 15.0 |
| median ratio | 8.1 | 10.5 | 8.5 | 5.4 | 17.9 | 6.4 | 10.3 |





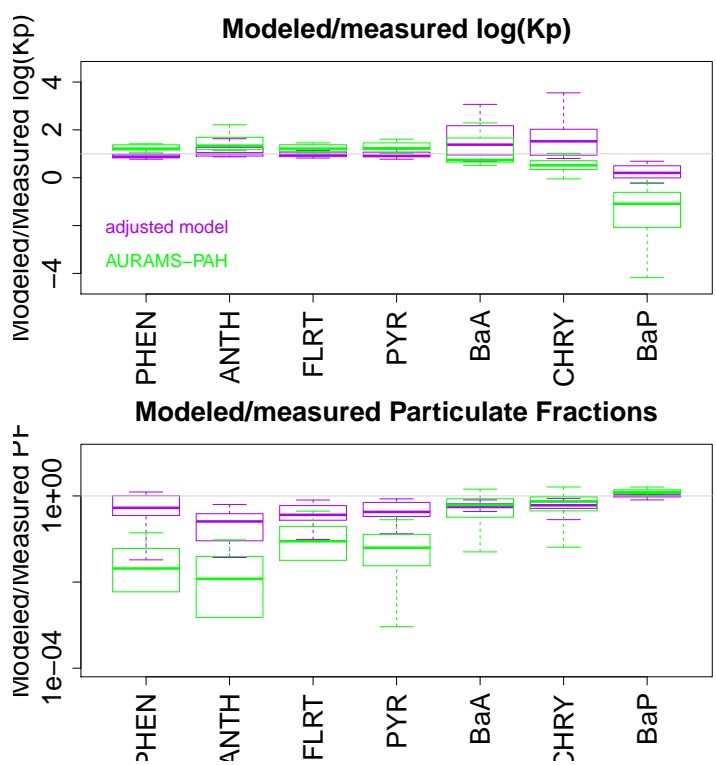

**Figure 2.** All-site ensemble of modeled/measured ratios of (top) log Kp, and (bottom) particulate fraction, for each PAH. Shown is the "adjusted model" (purple, from Eq. (B.2.1)), and the original AURAMS-PAH model (green). Box and Whiskers: thick line is the median, boxes extend to the $25^{th}$ and $75^{th}$ percentiles, and whiskers extend to the minimum and maximum.





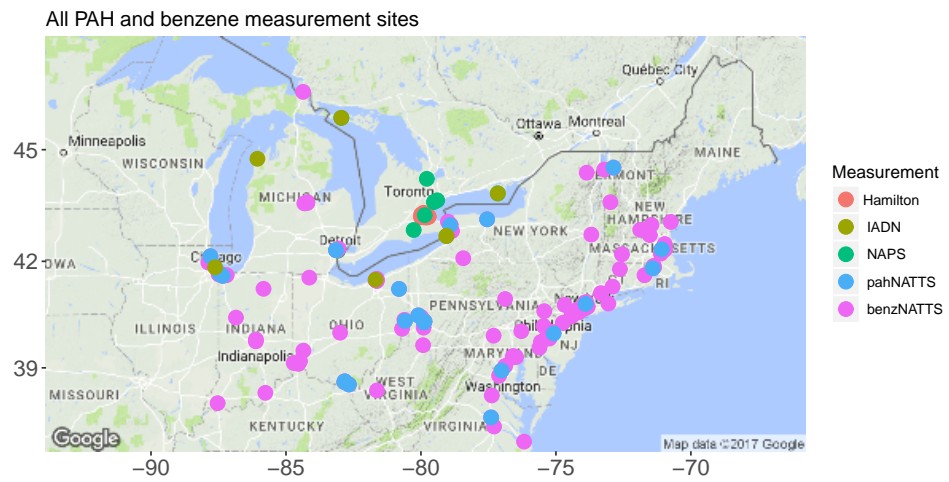

**Figure 3.** NAPS, NATTS, IADN and Hamilton measurement sites in the model domain.



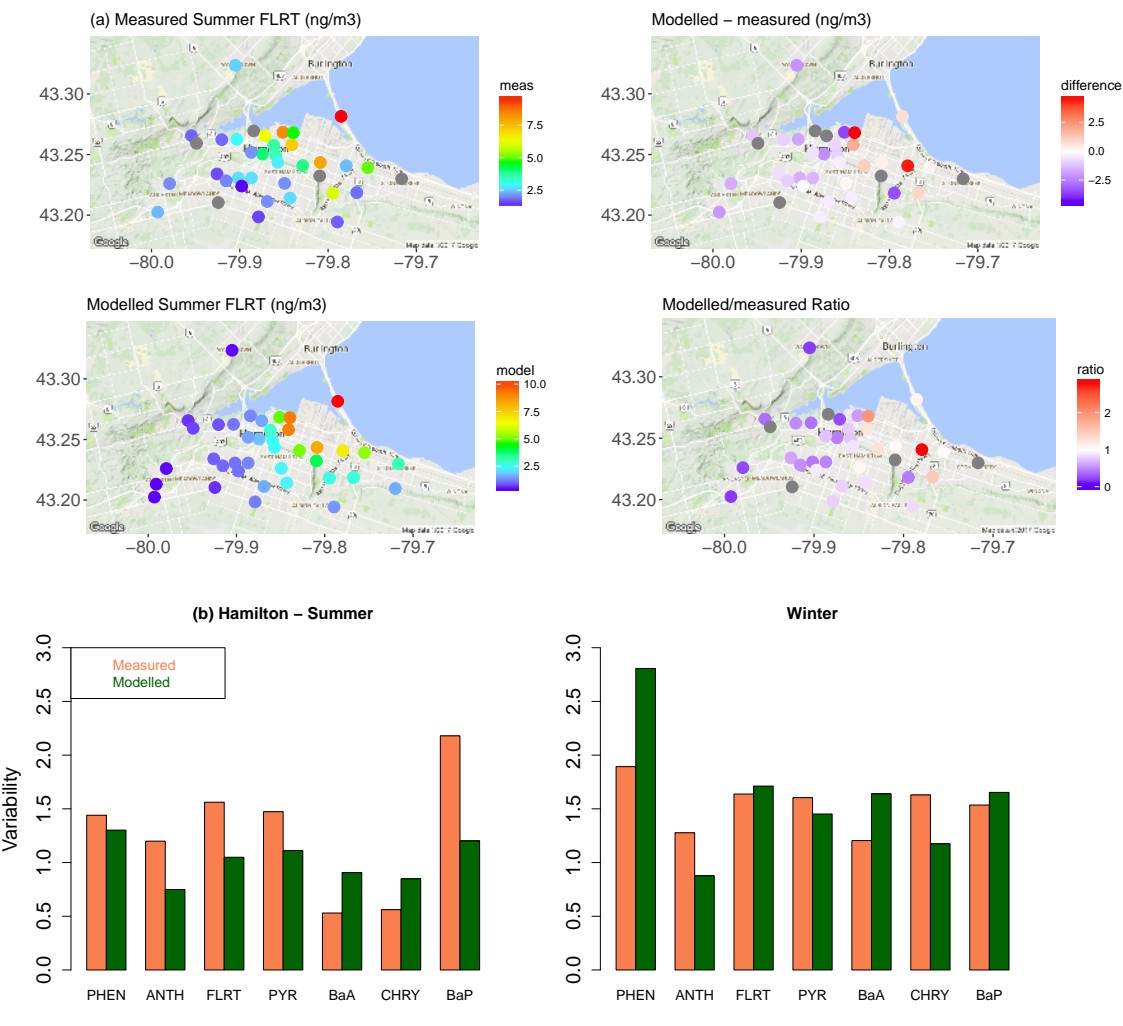

**Figure 4.** (a) Map of 2-week summertime average fluoranthene concentrations in Hamilton, Ontario: (left) from measurements and GEM-MACH-PAH model, and (right) their differences and ratios. (b) Spatial variability in the Hamilton data in (left) summer, and (right) winter.

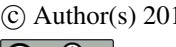

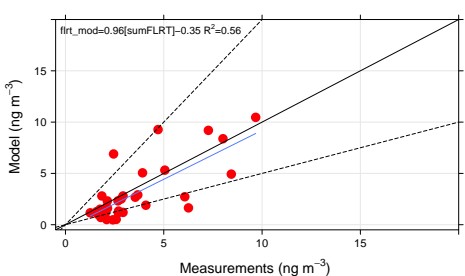

(a) Hamilton Summer Model vs Measurements – FLRT

**(b) Model/measurement concentration ratios for Hamilton**

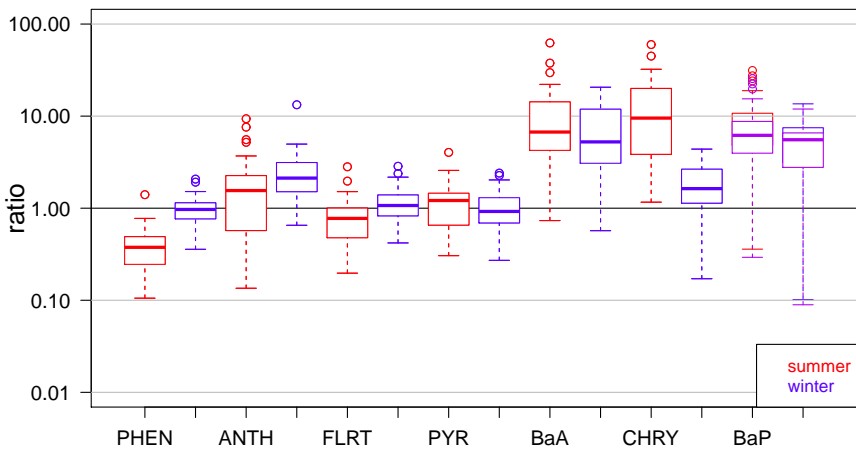

**Figure 5.** (a) GEM-MACH-PAH model vs. measurement scatter-plot of 2-week summertime fluoranthene concentrations at 40+ sites in Hamilton. (b) Frequency distributions of GEM-MACH-PAH model/measurement ratios of PAH concentrations for the Hamilton measurement-model pairs for all sites from both summer and winter. Purple boxes are results from $O_3$-corrected BaP measurements.



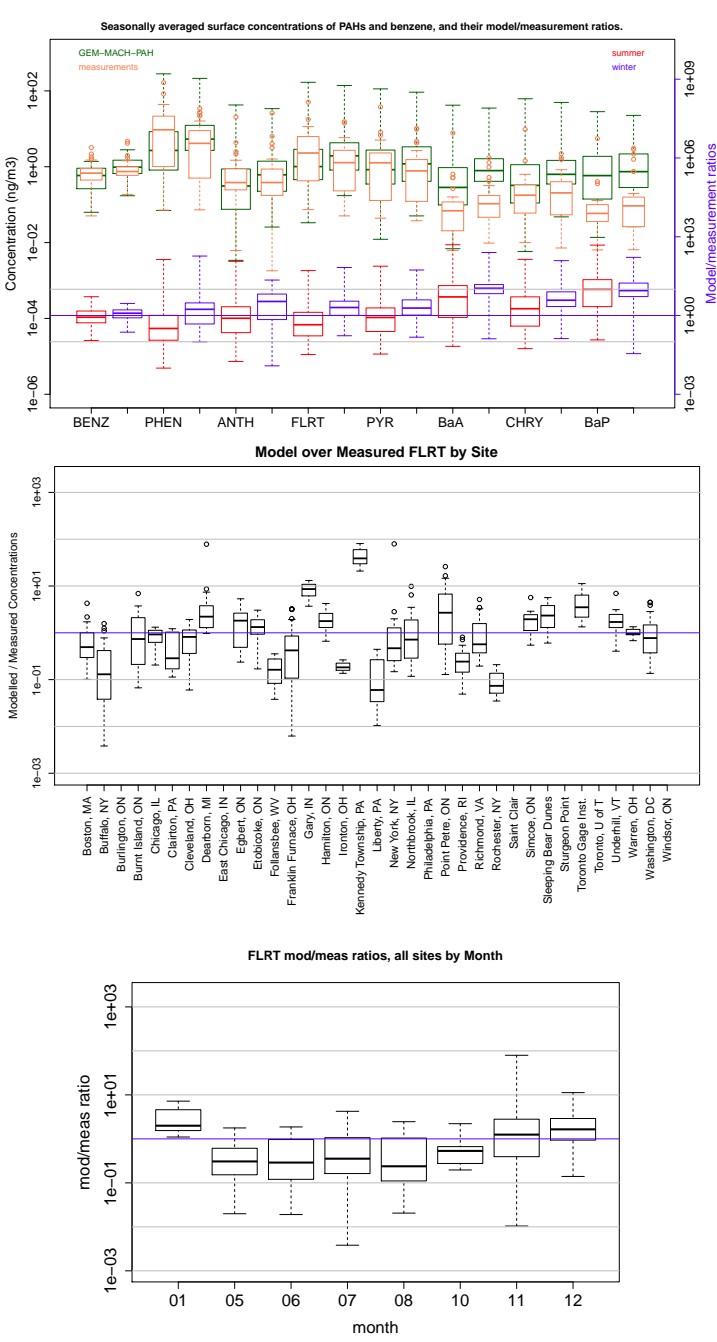

**Figure 6.** (a) Frequency distributions of GEM-MACH-PAH (green) and measured (orange) benzene (gas) and PAH (gas+particle) seasonal average concentrations at all IADN, NAPS, and NATTS sites. Modelled/measured concentration ratios also shown for summer (red) and winter (blue), with grey lines indicating agreement within an order of magnitude. (b) Modelled/measured concentrations for each daily model-measurement pair, separated by site (FLRT given as example), (c) same as (b) but separated by month.



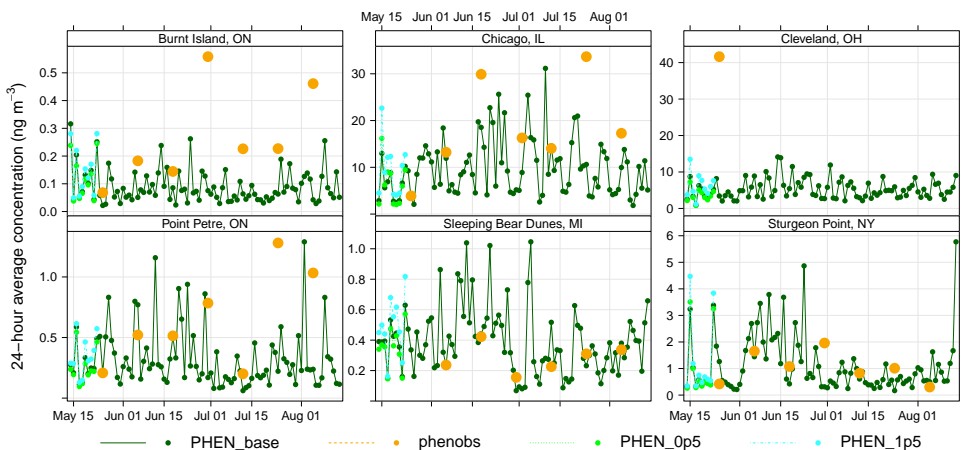

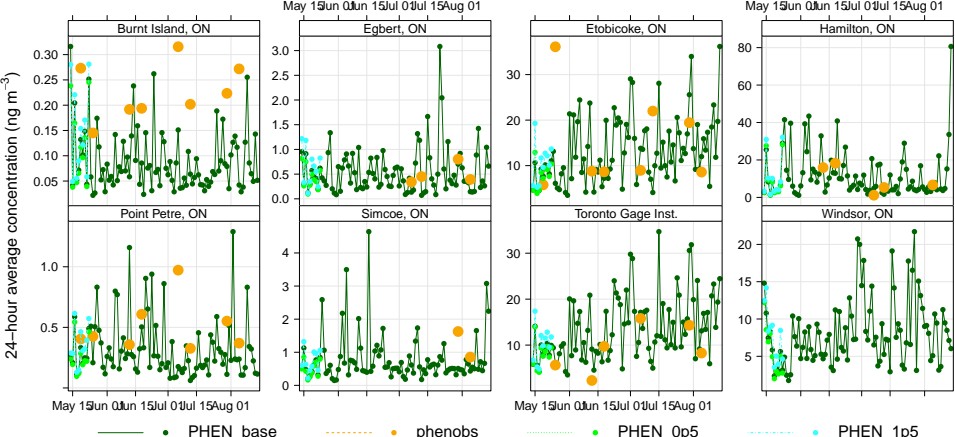

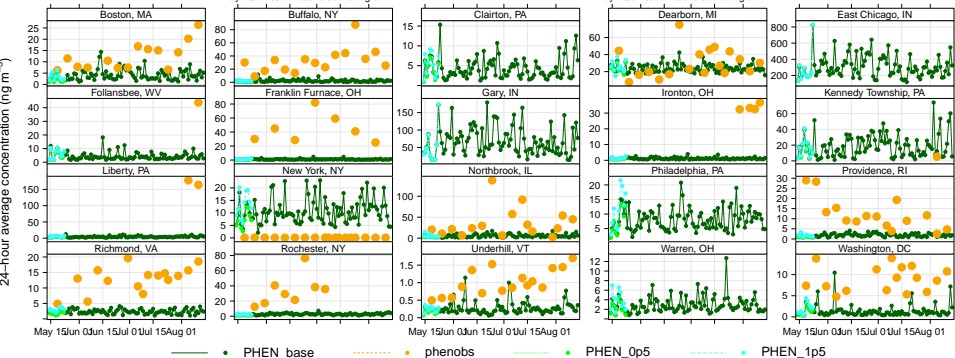

**Figure 7.** Phenanthrene time series for the summer 2009 period for the IADN (binational), NAPS (Canada) and NATTS (U.S.) networks. Orange=measurements, dark green=base GEM-MACH-PAH model, light green=GEM-MACH-PAH with 0.5×mobile emissions, cyan=GEM-MACH-PAH with 1.5×mobile emissions.





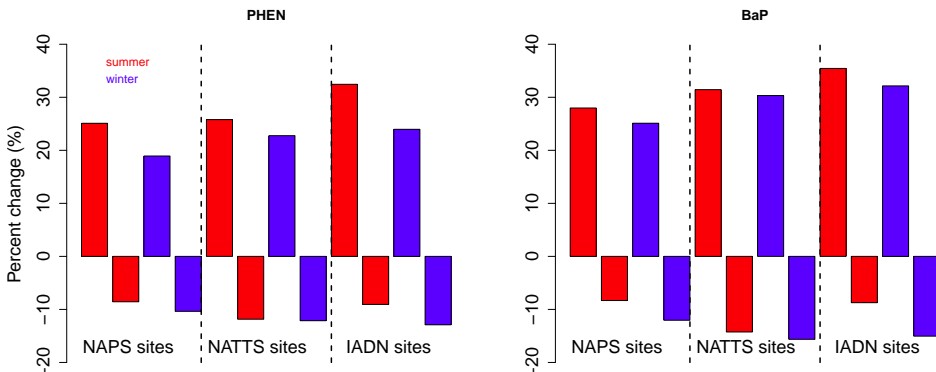

**Figure 8.** Average percent change in surface PHEN and BaP concentrations by season when PAH on-road mobile emissions are scaled up or down by factors of 1.5 and 0.5, respectively.

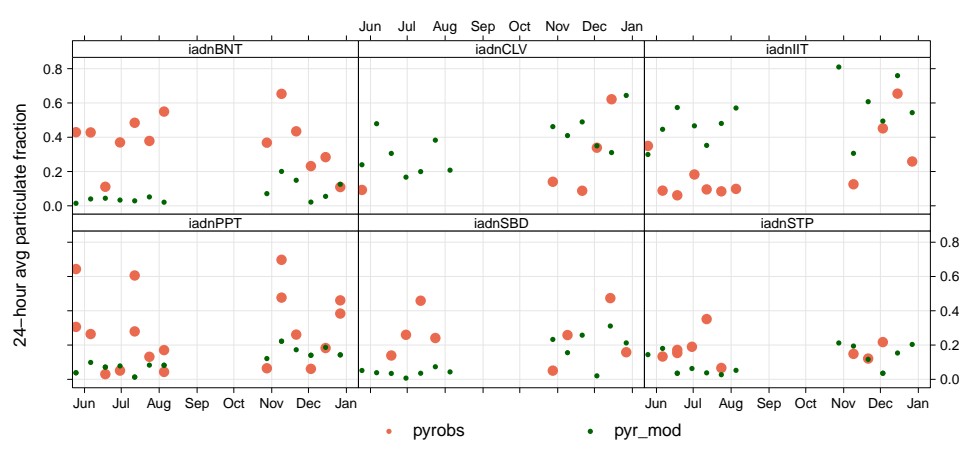

**Figure 9.** Time series of pyrene particulate fraction at six IADN network sites (BNT=Burnt Island, CLV=Cleveland, IIT=Chicago, PPT=Point Petre, SBD=Sleeping Bear Dunes, and STP=Sturgeon Point). GEM-MACH-PAH values are denoted by (green dots) and IADN measurements by orange dots.





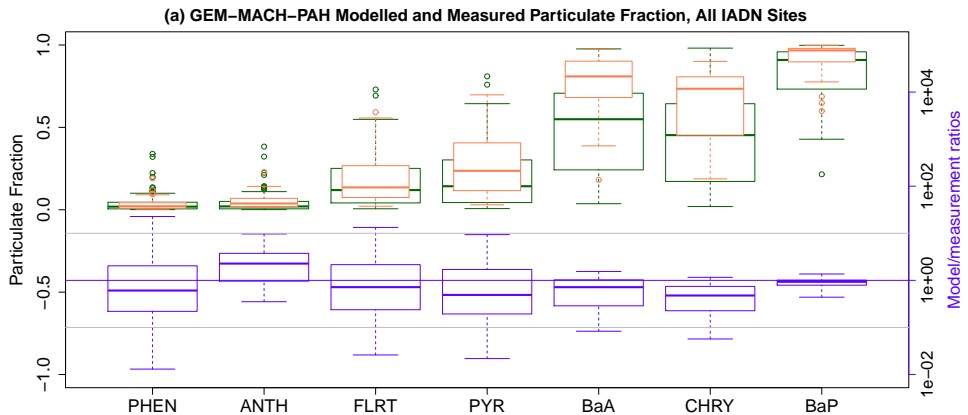

**Figure 10.** (a) GEM-MACH-PAH modelled (green) and measured (orange) particulate fraction ($\phi$) of all PAHs at all IADN sites, and their model/measurement ratios. (b) Same as (a), but for partitioning coefficient (log $K_p$). The blue line indicates the 1-to-1 line, and the gray lines are for ratios of 10 and 0.1.



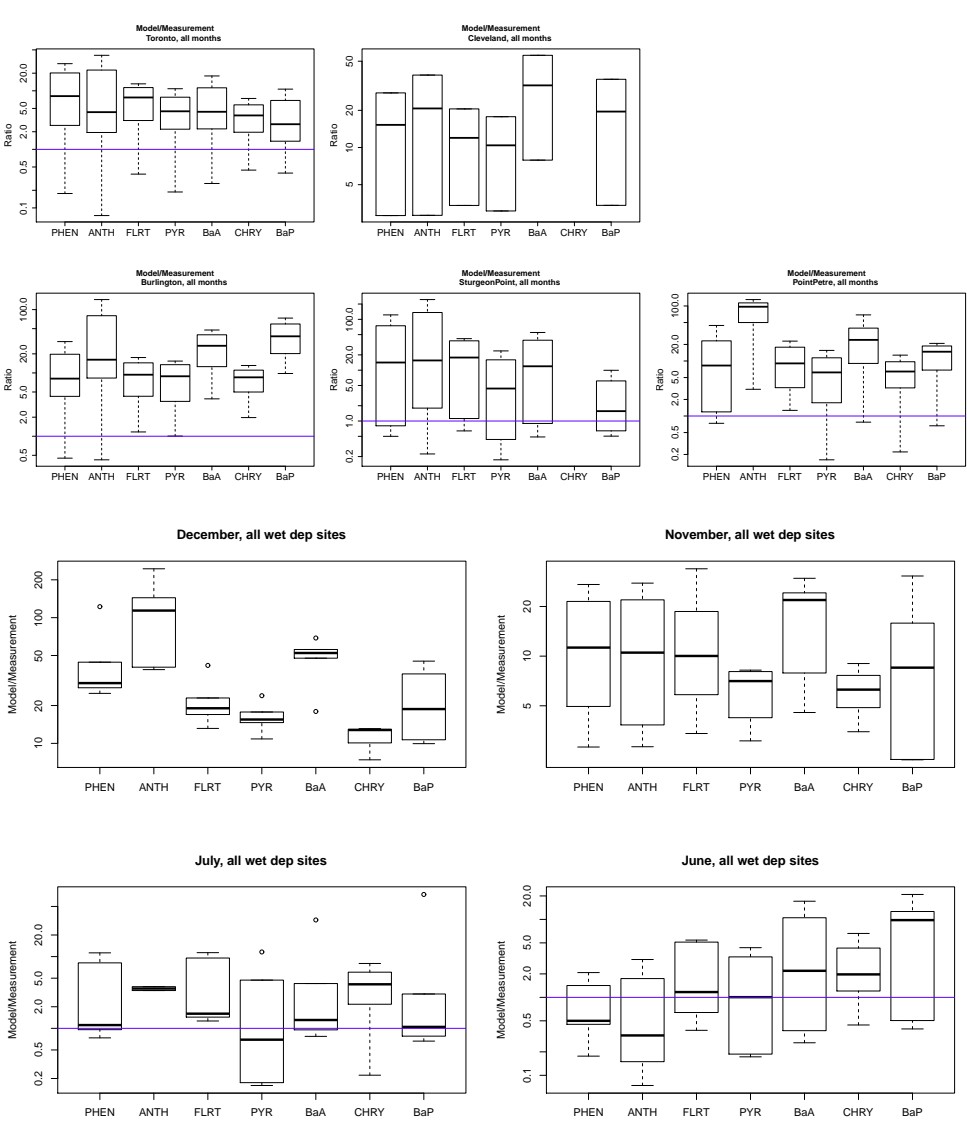

**Figure 11.** GEM-MACH-PAH model/measurement wet deposition ratios for all PAHs a) for five sites (all months) and b) for four months (all IADN sites).





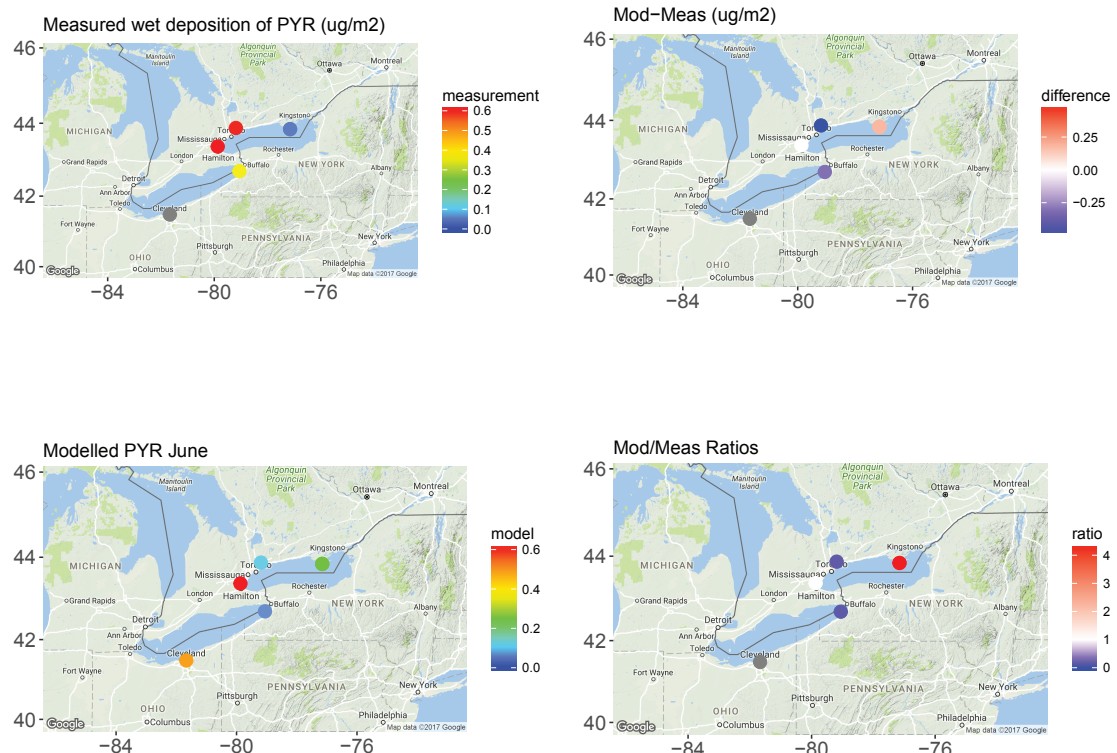

**Figure 12.** One-month (June 2009) wet deposition of pyrene from the (left) measurements and GEM-MACH-PAH model and (right) their differences and ratios.