# Peer review of "GEM-MACH-PAH (rev2488): a new high-resolution chemical transport model for North American PAHs and benzene"

_Geoscientific Model Development, 2017_

## Referee Comment (RC1) · Anonymous Referee #1 · 26 Mar 2018

General comments:

This discussion paper addresses the high-spatial resolution modelling of PAHs, which is relevant within the scope of GMD and important for the estimation of air quality. The presented model improves upon existing PAH prediction capabilities, making novel improvements to an existing tool's performance. The methods and assumptions are generally clearly outlined and valid with the conclusions supported by the results, with the exception of some specifics discussed below. The specific model improvements discussed are precisely and clearly presented, and therefore should be reproducible. To the best of my knowledge, the authors give proper credit to related work and clearly

indicate their own contribution. The title clearly reflects the contents of the paper and the abstract provides a complete and relatively concise summary. Overall, the paper is very well structured and clear, and the language is completely fluent and precise. The number and quality of the references is appropriate, as is the supplementary material. The authors provide a link to model source code, but do not include a user manual or compilation/run instructions and dependencies.

This paper represents an advance in PAH modelling, where the model appears to be limited by the availability of inputs, particularly emissions; the inability of such a high-resolution model to capture daily variability appears to be likely due to unresolved variability in emissions. However, the ability to make even seasonal-scale PAH estimates comparable to highly local measurements given emissions scenarios is an important asset for air-quality science.

While the abstract, conclusion, and body of the paper include ambiguous use of "statistically indistinguishable" which overstates the performance of model, the actual performance of the model represents a sufficient advance for PAH modelling.

––––––––––––––––––––––––––––––––––––––––––––––––––––––––––––––––––––––––––

Specific comments:

Adjusting Ksw to measurements taken in 2002 is a good way to navigate their high uncertainty, but some of the adjustments are incredibly large. E.g. PHEN and PYR Ksw increases by almost 2 orders of magnitude. The authors should discuss the justification of such a large change in the context of the prior uncertainty and/or possible missing mechanisms.

Equation 6 holds only if m_i_gas is equal to the total mass of PAH in the parcel of air considered; i.e. prior to partitioning, all of the PAH is gas-phase. Does this mean that once the PAH partitions to water it is considered lost? Is partitioning to cloud-water irreversible? A clarification of the fate of PAH that undergoes water uptake but not precipitative loss from the atmosphere is warranted.

[Figure]

ca. L460 Biases of BaA and BaP indicate NO3 reactions that are noted in the literature. Here a few more lines of discussion of this point would be helpful. A back-of-the envelope estimate of the effect of these reactions would increase confidence that these are the reason (or not) for the remaining bias.

ca. L560 The R-values in these site-by-site comparisons are fairly low even after the anomalous sites are removed. Discussion of the causes of this low correlation is warranted. At the considered time and spatial scales, is unresolved time-variability in emissions too large? The conclusions should highlight further the reliance on accurate emissions and their time and spatial distributions for daily and small-scale predictions.

The inability of the model to reproduce short-timescale PAH concentrations is a weakness considering its high resolution.

The number of aspects of the simulation that are compared to observations is a major asset for this work. The comparison to Kp, particulate fraction, wet deposition, and concentrations across many sites and PAHs allows a very detailed and transparent assessment of the model.

The winter/summer differences in wet deposition show that snow-initiated wet deposition is a definite weakness of the model. In the conclusions, while the authors mention that snow scavenging is new to the model, it should be acknowledged that it requires improvements going forward, along with possibilities of what these improvements might be.

ca. L675 "but it is at least promising to see that there are no particular sites where the model is consistently too high or too low, rather the errors in spatial distribution are haphazard and may be due to propagation of error, rather than any major error with the PAH scavenging scheme itself" I think that this paragraph should be re-written. The difference in the model compared to the observations is definitely due to the propagation of error, in all cases. The fact that these errors are different for different PAHs and sites is a result of the complexity of the processes involved, as the authors write, but I

do not think that this makes them more or less promising.

ca. L705 "we have determined from our sensitivity test that the GEM-MACH-PAH model has a linear response to a 50% variation in mobile emission factors, simulating concentrations that vary up to 30%." In the results section, the authors present a /non-linear/ response of concentrations to variations in mobile emission factors. Increasing emission from 50% to 100% yields ∼10% difference in concentrations, while increasing from 100% to 150% yields ∼30% difference in concentrations.

Figure 8) Further displays non-linearity of mobile source factor. What is the reason that it is not linear? What change in total emissions results from the mobile source change?

Abstract, L694 and ca. L570 "with concentrations statistically indistinguishable from observations, at 2.5-km resolution". If I understand the analysis, this phrasing highly overstates the performance of the model. Firstly, the 2.5 km resolution is not a significant part of the model-measurement comparison. The sites are a few dozen distributed all across the Northeastern U.S. and Southern Ontario, and are aggregated, and therefore the comparison is not testing the high spatial resolution. Secondly, by grouping all of the measurement-model pairs for the whole domain and season, a more accurate statement of the agreement would be "Over the domain as a whole and on the seasonal time-scale, the model is unbiased with respect to measurements." The above phrasing is misleading and ambiguous, and must be changed to at least clearly state the statistical test performed.

––––––––––––––––––––––––––––––––––––––––––––––––––––––––––––––––––––––––––––

Technical corrections:

Equation 1: "b" in equation, but "B" in text

L165 "amoung" should be "among"

L222 "In order to investigate whether these U.S. values would be representative of conditions in Canada and whether only have those two fuel-type categories are adequate,

. . ." I believe should read: "In order to investigate whether these U.S. values would be representative of conditions in Canada and whether having only these two fuel-type categories is adequate, . . ."

Equation 4: the "reduced" and "i" labeling is confusing; indicating a forward time-stepping would be clearer.

L269 ". . .for the seven PAHs is a linear relationship with inverse temperature." Should read ". . .for the seven PAHs are proportional to inverse temperature."

L380 "NATTS is an a U.S." should read "NATTS is a U.S."

L444 ". . . slope of the best-fit line is very close to 1." Here it is preferable to simply quantify the slope and remove the qualitative phrase "very close".

L648 "gages" should read "gauges"

Figure 4: The white circles are difficult to see on difference maps. A more visible color should be used. In all 4 map panels, the grey color of some of the dots is not on the color scale.

Figure 5 a) text too small Figure 5 b) great figure, but purple overlay hard to discern.

Figure 6 a) Should remove irrelevant labels on each y-axis (<=1e4 on left, >1e3 on right)

Figure 10 a) remove meaningless negative particle fraction axis labels Figure 10: b) does not exist but is mentioned in Figure 10 caption.

---

## Author Comment (AC1) · 8 May 2018

**Authors' response to reviewer 1**
Itallic font is quoted from reviewer.
Author responses begin with [CW].

**Anonymous Referee 1** General comments: This discussion paper addresses the high-spatial resolution modelling of PAHs, which is relevant within the scope of GMD and important for the estimation of air quality. The presented model improves upon existing PAH prediction capabilities, making novel improvements to an existing tool's performance. The methods and assumptions are*

[Figure]

*generally clearly outlined and valid with the conclusions supported by the results, with the exception of some specifics discussed below. The specific model improvements discussed are precisely and clearly presented, and therefore should be reproducible. To the best of my knowledge, the authors give proper credit to related work and clearly indicate their own contribution. The title clearly reflects the contents of the paper and the abstract provides a complete and relatively concise summary. Overall, the paper is very well structured and clear, and the language is completely fluent and precise. The number and quality of the references is appropriate, as is the supplementary material.*

*The authors provide a link to model source code, but do not include a user manual or compilation/run instructions and dependencies. This paper represents an advance in PAH modelling, where the model appears to be limited by the availability of inputs, particularly emissions; the inability of such a highresolution model to capture daily variability appears to be likely due to unresolved variability in emissions. However, the ability to make even seasonal-scale PAH estimates comparable to highly local measurements given emissions scenarios is an important asset for air-quality science. While the abstract, conclusion, and body of the paper include ambiguous use of "statistically indistinguishable" which overstates the performance of model, the actual performance of the model represents a sufficient advance for PAH modelling.*

[CW] Thank you for your thorough review of our paper. We have taken all of your comments into account in our revised manuscript. Please see below for more details. Regarding the overstatement of the model's performance: please our responses to the specific comments below.

Regarding the source code: while the code is in the public domain (the Zenodo site), unfortunately ECCC does not have the resources to support a community model, and therefore does not have instructions available for the compilation of the code and its underlying subroutine libraries on various platforms. However, I had added the following information to our Zenodo record:

"GEM-MACH is an extension of the standard GEM model which is available from https://github.com/mfvalin?tab=repositories. The executable for GEM-MACH is obtained by providing this chemistry library to GEM when generating its executable."

In addition, I've added a figure of the calling sequence for the model to the revised supplemental material. The "Data and code availability" section of the revised paper has been updated with this additional information. I hope this is sufficient.

**Specific comments:**

*Adjusting Ksw to measurements taken in 2002 is a good way to navigate their high uncertainty, but some of the adjustments are incredibly large. E.g. PHEN and PYR Ksw increases by almost 2 orders of magnitude. The authors should discuss the justification of such a large change in the context of the prior uncertainty and/or possible missing mechanisms.*

[CW] Firstly, your comment helped us to notice that the Ksw values we had in Table 1 of the manuscript were actually values that we initially calculated before we followed through with the method described in the supplemental material, starting at the bottom of p9, continuing top of p.10 (related to selecting samples that contained BaP rather than all of the samples, many of which were missing BaP measurements). Table 1 in the revised manuscript is now updated to the correct values used, but these are not much different from those in the original manuscript – thus your comment still applies. However, the Ksw values we calculated in this study – while they are different from the Galarneau et al (2014) values – are still in line with Ksw values found in published sources. See Table R.1. below, and note that the Dachs Eisenreich (2000) values for phenanthrene are of the same order of magnitude as our empirically-derived Ksw. There is also a range from Jonker and Koelmans (2002), which for anthracene reach close to our value. Our value for pyrene is not too much greater than the top end found in Bucheli Gustafsson (2000, EST), and so on. We have added the quoted literature sources of Ksw to the Table 1 in the revised manuscript.

Table R.1: KSW from the literature, and this study.

Putting in context of the prior uncertainty and/or possible missing mechanisms: We use Ksw combined with Kaw to estimate Ksa. This estimation may be inaccurate if equilibrium is not achieved. There is also no temperature dependence in published Ksw (but there is in Kaw) which can make a large difference at some temperatures and will be noticeable for compounds whose partitioning varies a lot across the gas-to-particle spectrum depending on temperature. The large correction and large range of published Ksw values lends further support for the need to measure Ksa directly on soots of atmospheric relevance. We have added this extra information into the revised manuscript.

*Equation 6 holds only if $m_{igas}$ is equal to the total mass of PAH in the parcel of air considered; i.e. prior to partitioning, all of the PAH is gas-phase. Does this mean that once the PAH partitions to water it is considered lost? Is partitioning to cloud-water irreversible? A clarification of the fate of PAH that undergoes water uptake but not precipitative loss from the atmosphere is warranted.*

[CW] In GEM-MACH we have separate tracers for particulate PAHs and gas-phase PAHs which are passed into the scavenging subroutines. There could be particle-phase PAHs in the same parcel, which would undergo scavenging via the particle-scavenging mechanism. However, if we understand the reviewer's comment correctly, we think they are talking about the gas-water partitioning of Equation 6, which applies only to the gas-phase mass of PAHs in the parcel. Yes, at the start of each chemistry time step, all of the gaseous PAHs are considered to be in the gas phase (none start out in the aqueous phase, and the within-cloud aqueous fraction is calculated at every time step when cloud is present). To further clarify the cloud-water scavenging process: Once the gaseous PAHs partition to cloud-water in GEM-MACH (which is re-calculated at each time step) they are subject to rain-out (cloud-to-rain conversion) process; the relative amount of PAH mass within the cloud water which is transferred to precipitation is thus lost from the amount within the cloud. At the end of each chemistry step,
the fraction of the dissolved PAH tracers contained within cloud water which are not removed by this rain-out process will be returned to gas phase. For the fraction that go into rain water, there is also a parameterization in the rain scavenging code for possible evaporation of rain before it reaches the ground. That fraction of PAHs released via evaporation of precipitation would also return to the atmosphere, albeit in the column below the cloud. The remaining fraction that isn't evaporated is counted towards the wet deposition in the model, which is treated as irreversible (i.e., no re-emission of PAHs from the ground after they are deposited). The revised manuscript has been updated with these further clarifications, and we hope they satisfy the reviewer's question.

*L460 Biases of BaA and BaP indicate NO3 reactions that are noted in the literature. Here a few more lines of discussion of this point would be helpful. A back-of-the envelope estimate of the effect of these reactions would increase confidence that these are the reason (or not) for the remaining bias.*

[CW] According to Keyte et al (2013), the PAH-NO3 reaction is actually more important for gas-phase PAHs. Since BaA and BaP have a relatively small fraction in the gas-phase, a gaseous NO3 loss mechanism would have a relatively small impact on total (gas+particle) BaA and BaP. That said, Liu et al (2012, EST) determined kNO3 second order rate coefficients for heterogenous (on-particle) reactions with pyrene, chrysene, and benz(a)anthracene. The BaA reaction rate was kNO3 = 4.3E-12 cm3/molec/s. When we do a back-of-the-envellope calculation of BaA lifetime (the time required for BaA concentrations to drop to 1/e of its initial concentration) with respect to heterogenous reaction with NO3, using typical night-time concentrations of NO3 from the model ( 3.2pptv or 6.67E7 molecules/cm3), we get a lifetime of about 1 hour for particulate BaA. Thus, at moderate to high concentrations of NO3, gas-phase reactions of PAHs with NO3 may be a significant loss, and thus a probable cause for the remaining bias. However, this calculation assumes that the one Liu et al (2012) study is correct and applies to all atmospheric conditions (temperatures, etc). More research should be

done to determine whether this reaction should go into PAH CTMs. We have added this discussion to the revised manuscript at the end of that paragraph.

*L560 The R-values in these site-by-site comparisons are fairly low even after the anomalous sites are removed. Discussion of the causes of this low correlation is warranted. At the considered time and spatial scales, is unresolved time-variability in emissions too large? The conclusions should highlight further the reliance on accurate emissions and their time and spatial distributions for daily and small-scale predictions. The inability of the model to reproduce short-timescale PAH concentrations is a weakness considering its high resolution.*

[CW] We agree that accurate emissions at finer time and spatial resolutions would greatly improve model results. Ideally, increasing model resolution would increase R at shorter timescales, however, given the large dynamic range and sharp spatial gradients that PAH concentrations have and the difficulty in modelling plume locations accurately, in reality we end up with lower R values at higher resolution than those reported in other studies*. Thus, the errors in modelled transport are likely another cause for the low correlation coefficients at high time and spatial resolution (in addition to the unresolved time-variability in emissions). We have added this to the text of the revised manuscript.

It was not our intention to dwell on the R values in the model-measurement comparison because of the high uncertainties in PAH modelling and emissions. For example, we did not report on the R values we obtained for the NAPS network analysis (which were much better than those for the NATTS network, at about R=0.6-0.7 depending on the PAH species). However, we mentioned them in this part of the text to emphasize the improvement when gridpoints with known point-source emission errors are removed.

*Note many other PAH modelling studies do not report correlation coefficients (e.g., Aulinger et al, 2011 reported index of agreement instead of R; Zhang et al, 2016, reported mean fractional bias and mean fraction error), and those that do report R

values have model-measurement pairs that have been averaged over larger spatial scales or larger timescales (or both). For example, Aulinger et al (2007) achieved better R values of 0.75 for weekly averages, and 0.58 for 2-day averages, however, they only had 6 measurement sites in their study, and coarser grid (18-km) resolution, which may have smoothed over some of the variability we see at higher resolution. Also Friedman et al (2012) achieved good R values of 0.64 or higher, however, these were for their 5-year mean data over 15-16 sites globally (most sites in Europe, NE-North America, and China), at 4ox5o resolution – both spatial and time scales that smooth out variability. Matthias et al (2009) report good correlation coefficients of 0.3 to 0.8, but these were for the times series at each site, rather than for all sites together. Also their model was at coarser (1ox1o) resolution than ours.

*The number of aspects of the simulation that are compared to observations is a major asset for this work. The comparison to Kp, particulate fraction, wet deposition, and concentrations across many sites and PAHs allows a very detailed and transparent assessment of the model.*

[CW] Thank you!

*The winter/summer differences in wet deposition show that snow-initiated wet deposition is a definite weakness of the model. In the conclusions, while the authors mention that snow scavenging is new to the model, it should be acknowledged that it requires improvements going forward, along with possibilities of what these improvements might be.*

[CW] We added this text to the conclusion (at the end of the first paragraph): "However, the modelled wet deposition was biased high - particularly in the wintertime - thus further improvements to these parametrizations are required if the model is to be used for deposition studies." and we added some additional text to the first paragraph of section 4.4: "…removing PAHs in the model. This may be due to our simplification of a constant snow surface area, which may be set too high, or due to inaccuracies in the

modelled or measured precipitation."

The simplification of a constant snow surface area may have introduced a high bias in snow scavenging if actual surface area of snow was smaller. Ranges of 0.01 m2/g to 15 m2/g have been reported in the literature for snow surface area (e.g., Hoff et al, 1998; Hanot et al, 1999; Legagneux et al, 2002; Domine et al, 2007; and Hachikubu et al, 2014), but these are based usually on measurements from fallen snow (rather than falling snow). Fresh snow surface area was usually on the higher end, which is why we chose a value of 1 m2/g, which was at the high end of the reported values. The interfacial adsorption coefficient was dependent on modelled saturation vapour pressure for PAHs, which also may have introduced errors. The modelled gas-particle partitioning is likely not to blame, because the modelled particulate fraction was underestimated compared to IADN measurements (Fig.10) and snow scavenging of particles is more efficient than that of the gas-phase.

*L675 "but it is at least promising to see that there are no particular sites where the model is consistently too high or too low, rather the errors in spatial distribution are haphazard and may be due to propagation of error, rather than any major error with the PAH scavenging scheme itself" I think that this paragraph should be re-written. The difference in the model compared to the observations is definitely due to the propagation of error, in all cases. The fact that these errors are different for different PAHs and sites is a result of the complexity of the processes involved, as the authors write, but I do not think that this makes them more or less promising.*

[CW] We have rephrased the second half of that paragraph as follows: ". . . highest in Burlington). The lack of spatial or temporal pattern in the sign and/or magnitude of wet deposition biases indicates that there is no major error with the PAH scavenging scheme itself. Given that wet deposition of PAHs relies on the correct simulation of many model factors (meteorology, scavenging parameters, atmospheric concentrations, etc), our work suggests that more process studies aimed at quantifying wet deposition are needed. In fact, other PAH models also overestimate PAH deposition

(Matthias et al, 2009; Friedman et al, 2012)."

*L705 "we have determined from our sensitivity test that the GEM-MACH-PAH model has a linear response to a 50% variation in mobile emission factors, simulating concentrations that vary up to 30%." In the results section, the authors present a /non-linear/ response of concentrations to variations in mobile emission factors. Increasing emission from 50% to 100% yields 10% difference in concentrations, while increasing from 100% to 150% yields 30% difference in concentrations. Figure 8) Further displays non-linearity of mobile source factor. What is the reason that it is not linear? What change in total emissions results from the mobile source change?*

[CW] Our apologies, this is an important error in the manuscript that you caught. What we actually did was scale the mobile emissions by 0.5 (halved) and 2.0 (doubled, not 1.5). And what we get when we sum over the domain in total PAH area emissions (doesn't include major points) for each scaled scenario is in Table R.2 below – which was calculated for 14 UTC on a Monday in October as an example. This amounts to -10% for the 0.5x change in mobile emissions and +20% for the 2.0x change in mobile emissions. So the results are linear, as the latter is double the former. The manuscript has been corrected and clarified and all mention of 1.5x has been replaced with 2.0x.

Table R.2: Total PAH emissions (g/s) in the Pan Am domain for 14 UTC on Monday in October

*Abstract, L694 and ca. L570 "with concentrations statistically indistinguishable from observations, at 2.5-km resolution". If I understand the analysis, this phrasing highly overstates the performance of the model. Firstly, the 2.5 km resolution is not a significant part of the model-measurement comparison. The sites are a few dozen distributed all across the Northeastern U.S. and Southern Ontario, and are aggregated, and therefore the comparison is not testing the high spatial resolution. Secondly, by grouping all of the measurement-model pairs for the whole domain and season, a more accurate statement of the agreement would be "Over the domain as a whole and on the sea-*

*sonal time-scale, the model is unbiased with respect to measurements." The above phrasing is misleading and ambiguous, and must be changed to at least clearly state the statistical test performed.*

[CW] We have rephrased the text in the revised manuscript to better indicate the model's performance as you've suggested. E.g., the conclusion now reads as follows: "Over the model domain, at seasonal time-scales, the GEM-MACH-PAH simulation of benzene and six semi-volatile PAHs (PHEN, ANTH, FLRT, PYR, BaA, CHRY) is statistically unbiased with respect to measurements (t<1 and p>0.05). For the seventh PAH species, BaP; its summertime average is simulated to a similar level of accuracy. However, it appears the model's OH, O3 and PM biases were additive, resulting in a wintertime average that is biased significantly high for BaP."

The abstract was similiarly changed. Finally, near line 570, when t<1 and p>0.05, this by definition means that the two datasets have no significant difference. Therefore, our expression that the model was indistinguishable from the measurements is technically true. However, to not overstate the performance of the model, we have added the caveat in the revised version that it is when taking all of the data across the domain (rather than at each specific site).

That said, our analysis of the Hamilton, ON measurement campaign has specifically assessed the model's spatial performance at high resolution.

**Technical corrections:**

*Equation 1: "b" in equation, but "B" in text*

[CW] Corrected in revised manuscript.

*L165 "amoung" should be "among"*

[CW] Corrected in revised manuscript.

*L222 "In order to investigate whether these U.S. values would be representative of con-*

ditions in Canada and whether only have those two fuel-type categories are adequate,
. . ." I believe should read: "In order to investigate whether these U.S. values would
be representative of conditions in Canada and whether having only these two fuel-type
categories is adequate, . . ."

[CW] Thank you. The phrase has been revised.

*Equation 4: the "reduced" and "i" labeling is confusing; indicating a forward timestepping would be clearer.*

[CW] Thank you. We've revised such that [BaP]i is now [BaP]t and [BaP]reduced is
now [BaP]t+dt.

*L269 ". . .for the seven PAHs is a linear relationship with inverse temperature." Should
read ". . .for the seven PAHs are proportional to inverse temperature."*

[CW] Yes, thank you. This has been revised.

*L380 "NATTS is an a U.S." should read "NATTS is a U.S."*

[CW] Corrected in revised manuscript.

*L444 ". . . slope of the best-fit line is very close to 1." Here it is preferable to simply
quantify the slope and remove the qualitative phrase "very close".*

[CW] We've changed this to "the slope of the best-fit line is 0.96."

*L648 "gages" should read "gauges"*

[CW] Thank you. This has been corrected in the revised manuscript.

*Figure 4: The white circles are difficult to see on difference maps. A more visible color
should be used. In all 4 map panels, the grey color of some of the dots is not on the
color scale.*

[CW] We have added text in the caption to explain that the grey dots are sites that are
missing measurements in the summer (but not in the winter). We also changed the

background to a darker colour in the two right-hand maps so that the white dots are more visible (see Fig. 3 below):

Fig 3: replacement for Fig 4a in the paper.

*Figure 5 a) text too small Figure 5 b) great figure, but purple overlay hard to discern.*

[CW] Fig 5.a was made larger, and the font is now more readable. The corrected-BaP box has been moved to be beside the uncorrected BaP box – see Fig 4 below for the revised figure.

Fig 4: replacement for Fig 5b in the paper.

*Figure 6 a) Should remove irrelevant labels on each y-axis (<=1e4 on left, >1e3 on right)*

[CW] Thank you for this suggestion. These have been removed.

*Figure 10 a) remove meaningless negative particle fraction axis labels Figure 10: b) does not exist but is mentioned in Figure 10 caption.*

[CW] The plots have been split up and each has an appropriate y-axis (no negative particulate fractions, and no ratios greater than 100) – see Fig. 5 below. We removed the reference to (a) and (b) in the caption. Thank you!

Fig 5: Replacement for Fig.10 in the manuscript. New caption is: "GEM-MACH-PAH modelled (green) and measured (orange) particulate fraction (f) of all PAHs at all IADN sites, and their model/measurement ratios (blue). The blue line indicates the 1-to-1 line, and the grey lines are for ratios of 10 and 0.1."

Please also note the supplement to this comment:
https://www.geosci-model-dev-discuss.net/gmd-2017-324/gmd-2017-324-AC1-supplement.pdf
**Table R.1:** $K_{SW}$ from the literature, and this study.

| | PHEN | ANTH | FLRT | PYR | BaA | CHRY | BaP |
|---|---|---|---|---|---|---|---|
| **Original** from Galarneau et al (2014) | $4.34 \times 10^5$ | $1.55 \times 10^6$ | $2.24 \times 10^6$ | $1.70 \times 10^6$ | $3.74 \times 10^7$ | $2.82 \times 10^7$ | $9.59 \times 10^7$ |
| From Dachs Eisenreich (2000, ES&T) | $1.26 \times 10^7$ | NA | $6.31 \times 10^7$ | $5.01 \times 10^7$ | NA | $3.16 \times 10^8$ | NA |
| From Jonker and Koerlmans (2002, ES&T) | $1.86 \times 10^5$ - $3.72 \times 10^6$ | $3.80 \times 10^5$ - $1.26 \times 10^7$ | $5.01 \times 10^5$ - $9.12 \times 10^6$ | $5.13 \times 10^5$ - $8.91 \times 10^6$ | $4.07 \times 10^6$ - $1.82 \times 10^8$ | $3.24 \times 10^6$ - $3.39 \times 10^8$ | $2.57 \times 10^7$ - $1.17 \times 10^9$ |
| From Xu et al. (2012, E&ES) | $4.79 \times 10^5$ - $5.13 \times 10^5$ | $1.29 \times 10^6$ - $1.91 \times 10^6$ | $1.86 \times 10^5$ - $3.72 \times 10^6$ | $3.09 \times 10^6$ - $4.07 \times 10^6$ | $6.31 \times 10^7$ - $7.94 \times 10^7$ | $3.89 \times 10^7$ - $5.75 \times 10^7$ | $2.00 \times 10^8$ - $3.39 \times 10^8$ |
| From Bucheli and Gustafsson (2000, ES&T) | $2.57 \times 10^5$ - $5.89 \times 10^6$ | NA | $2.82 \times 10^5$ - $1.62 \times 10^6$ | $2.75 \times 10^6$ - $1.55 \times 10^7$ | NA | NA | NA |
| **this study** | $2.95 \times 10^7$ | $3.75 \times 10^7$ | $7.71 \times 10^7$ | $8.98 \times 10^7$ | $1.39 \times 10^8$ | $2.09 \times 10^7$ | $1.04 \times 10^8$ |

**Fig. 1.**

**Table R.2:** Total PAH emissions (g/s) in the Pan Am domain for 14 UTC on Monday in October

|              | 0.5x mobile emiss | 1.0x mobile emiss | 2.0x mobile emiss |
|--------------|-------------------|-------------------|-------------------|
| benzene      | 2530.3            | 3030.8            | 4031.7            |
| phen         | 25.3              | 29.0              | 36.5              |
| anth         | 3.5               | 3.8               | 4.6               |
| flrt         | 11.5              | 12.17             | 13.6              |
| pyr          | 11.15             | 11.9              | 13.5              |
| baa          | 4.3               | 4.7               | 5.4               |
| chry         | 4.13              | 4.5               | 5.11              |
| bap          | 5.0               | 5.8               | 7.4               |
| Total of 7 PAHs | 64.8           | 71.9              | 86.1              |

**Fig. 2.**

[Figure]

**Fig. 3.**

**(b) Model/measurement concentration ratios for Hamilton**

[Figure: box plot showing model/measurement concentration ratios for Hamilton, with y-axis "ratio" on a logarithmic scale from 0.01 to 100.00, and x-axis categories PHEN, ANTH, FLRT, PYR, BaA, CHRY, BaP, corBaP. Red boxes represent summer and blue boxes represent winter.]

summer
winter

**Fig. 4.**

[Figure]

**Fig. 5.**

**Supplement:**

GEM-MACH-PAH: a new high-resolution chemistry transport model for North American PAHs and benzene

Cynthia H. Whaley, Elisabeth Galarneau, Paul A. Makar, Ayodeji Akingunola, Wanmin Gong, Sylvie Gravel, Michael D. Moran, Craig Stroud, Junhua Zhang, and Qiong Zheng

SUPPLEMENTAL MATERIAL

CONTENTS

**Appendix A: Table of measurement sites**

**Table A.1:** Benzene and PAH network measurement sites in the Pam Am model domain. The following codes are used in the "measurement" column: "PB"=PAHs and benzene measured; "justP"=only PAHs measured; "justB"=only benzene measured; and "wetdep"=wet deposition measured.

Note that NATTS have site IDs, but not names, therefore the first column provides the state where the particular site resides, and cities/towns were manually looked up and added for those sites that have PAH measurements.

| Site name or State | Site ID | Latitude | Longitude | measurement |
|---|---|---|---|---|
| **IADN sites** | | | | |
| University of Toronto, ON | UOT | 43.8725 | -79.18833 | wetdep |
| St Clair, ON | STC | 42.53594 | -82.38978 | wetdep |
| Burlington, ON | BUR | 43.36889 | -79.87028 | wetdep |
| Burnt Island, ON | BNT | 45.82833 | -82.94806 | justP |
| Point Petre, ON | PPT | 43.84278 | -77.15361 | justP/wetdep |
| Sturgeon Point, NY | STP | 42.69306 | -79.055 | justP/wetdep |
| Cleveland, OH | CLV | 41.49214 | -81.67853 | justP/wetdep |
| Sleeping Bear Dunes, MI | SBD | 44.76111 | -86.0586 | justP |
| Chicago IIT, IL | IIT | 41.83444 | -87.6247 | justP |
| **NAPS sites** | | | | |
| College and South, Windsor, ON | 60211 | 42.29289 | -83.0731 | PB |
| Gage Institute, Toronto, ON | 60427 | 43.65822 | -79.3972 | PB |
| Etobicoke South (Toronto Kipling), ON | 60435 | 43.61076 | -79.5219 | PB |
| Elgin and Kelly, Hamilton, ON | 60512 | 43.25778 | -79.8617 | PB |
| Experimental Farm, Simcoe, ON | 62601 | 42.8569 | -80.2703 | PB |
| Egbert, ON | 64401 | 44.23111 | -79.7831 | PB |
| Point Petre, ON | 64601 | 43.84278 | -77.1536 | PB |
| Burnt Island, ON | 65501 | 45.82833 | -82.9481 | PB |
| **NATTS sites** | | | | |
| Middletown, OH | 390170003 | 39.4938 | -84.3543 | justB |
| OH | 390350038 | 41.47701 | -81.6824 | justB |
| OH | 390350068 | 41.45478 | -81.6344 | justB |
| OH | 390350069 | 41.519 | -81.6378 | justB |
| OH | 390350071 | 41.49251 | -81.67 | justB |
| OH | 390490034 | 40.00274 | -82.9944 | justB |
| OH | 390515502 | 41.55002 | -84.1365 | justB |
| OH | 390610014 | 39.19433 | -84.479 | justB |
| OH | 390610044 | 39.13837 | -84.7116 | justB |
| OH | 390610045 | 39.17093 | -84.5287 | justB |

| | | | | |
|---|---|---|---|---|
| OH | 390610046 | 39.11412 | -84.5363 | justB |
| OH | 390810017 | 40.36644 | -80.6156 | justB |
| Ironton, OH | 390875503 | 38.51814 | -82.6688 | PB |
| Franklin Furnace, OH | 391450020 | 38.60934 | -82.8225 | PB |
| Franklin Furnace, OH | 391450021 | 38.60066 | -82.8296 | PB |
| Franklin Furnace, OH | 391450022 | 38.58808 | -82.8348 | PB |
| Warren, OH | 391555504 | 41.23506 | -80.8127 | PB |
| MI | 260330901 | 46.49361 | -84.3642 | justB |
| MI | 261110951 | 43.60917 | -84.2106 | justB |
| MI | 261110953 | 43.59139 | -84.2094 | justB |
| MI | 261110955 | 43.58944 | -84.2211 | justB |
| MI | 261110959 | 43.57419 | -84.3216 | justB |
| MI | 261630015 | 42.30279 | -83.1065 | justB |
| Dearborn, MI | 261630033 | 42.30754 | -83.1496 | PB |
| MI | 261635502 | 42.35059 | -83.0524 | just B |
| Liberty, PA | 420030064 | 40.32377 | -79.8681 | PB |
| Clairton, PA | 420033007 | 40.29434 | -79.8853 | PB |
| Kennedy Township, PA | 420035503 | 40.49435 | -80.0964 | PB |
| PA | 420450002 | 39.83556 | -75.3725 | justB |
| PA | 420710007 | 40.04667 | -76.2833 | justB |
| PA | 420770004 | 40.61194 | -75.4325 | justB |
| PA | 420910005 | 40.19255 | -75.4575 | justB |
| PA | 421010004 | 40.00889 | -75.0978 | justB |
| PA | 421010014 | 40.04962 | -75.2408 | justB |
| PA | 421010047 | 39.94465 | -75.1652 | justB |
| PA | 421010055 | 39.92287 | -75.1869 | justB |
| PA | 421010063 | 39.88294 | -75.2197 | justB |
| PA | 421010136 | 39.9275 | -75.2228 | justB |
| PA | 421190001 | 40.95517 | -76.8819 | justB |
| PA | 421250005 | 40.14667 | -79.9022 | justB |
| PA | 420010001 | 39.92002 | -77.3097 | justB |
| PA | 420030031 | 40.44337 | -79.9903 | justB |
| Philadelphia, PA | 421010449 | 39.9825 | -75.0831 | justP |
| Buffalo, NY | 360291013 | 42.98844 | -78.9186 | PB |
| Rochester, NY | 360551007 | 43.1462 | -77.5481 | PB |
| NY | 360050133 | 40.8679 | -73.8781 | justB |
| NY | 360095501 | 42.08506 | -78.4336 | justB |
| NY | 360291007 | 42.7273 | -78.8498 | justB |
| NY | 360291014 | 42.99813 | -78.8993 | justB |
| NY | 360310003 | 44.39308 | -73.8589 | justB |
| NY | 360470118 | 40.69545 | -73.9277 | justB |
| NY | 360610115 | 43.84955 | -79.9357 | justB |

| | | | | |
|---|---|---|---|---|
| NY | 360632008 | 43.08218 | -79.0011 | justB |
| NY | 360810124 | 40.73614 | -73.8216 | justB |
| NY | 360831003 | 42.73194 | -73.6891 | justB |
| NY | 360850111 | 40.58027 | -74.1983 | justB |
| NY | 360850132 | 40.58056 | -74.1518 | justB |
| NY | 361030009 | 40.82799 | -73.0575 | justB |
| New York, NY | 360050110 | 40.81616 | -73.9021 | PB |
| CT | 90019003 | 41.11833 | -73.3367 | justB |
| CT | 90031003 | 41.78472 | -72.6317 | justB |
| CT | 90090027 | 41.3014 | -72.9029 | justB |
| DE | 100031008 | 39.5778 | -75.6107 | justB |
| DE | 100032004 | 39.73944 | -75.5581 | justB |
| NJ | 340155501 | 39.83709 | -75.244 | justB |
| NJ | 340210005 | 40.28309 | -74.7427 | justB |
| NJ | 340230006 | 40.47282 | -74.4224 | justB |
| NJ | 340230011 | 40.46218 | -74.4294 | justB |
| NJ | 340273001 | 40.78763 | -74.6763 | justB |
| NJ | 340390004 | 40.64144 | -74.2084 | justB |
| NJ | 340395502 | 40.65205 | -74.1999 | justB |
| MD | 240053001 | 39.31083 | -76.4744 | justB |
| MD | 240330030 | 39.05528 | -76.8783 | justB |
| MD | 245100006 | 39.34056 | -76.5822 | justB |
| MD | 245100040 | 39.29806 | -76.6047 | justB |
| Washington, DC | 110010043 | 38.92185 | -77.0132 | PB |
| Providence, RI | 440070022 | 41.80795 | -71.415 | PB |
| RI | 440030002 | 41.61524 | -71.72 | justB |
| RI | 440070026 | 41.87467 | -71.38 | justB |
| RI | 440071010 | 41.84157 | -71.3608 | justB |
| NH | 330110020 | 42.99578 | -71.4625 | justB |
| NH | 330111011 | 42.71866 | -71.5224 | justB |
| NH | 330115001 | 42.86175 | -71.8784 | justB |
| NH | 330150014 | 43.07533 | -70.748 | justB |
| MA | 250092006 | 42.47464 | -70.9708 | justB |
| MA | 250130008 | 42.19438 | -72.5551 | justB |
| MA | 250213003 | 42.21177 | -71.114 | justB |
| MA | 250250041 | 42.31737 | -70.9684 | justB |
| Boston, MA | 250250042 | 42.32944 | -71.0825 | PB |
| Northbrook, IL | 170314201 | 42.14 | -87.7992 | PB |
| O'Hare airport, Chicago, IL | 170313103 | 41.96519 | -87.8763 | justB |
| Underhill, VT | 500070007 | 44.52839 | -72.8688 | PB |
| VT | 500070014 | 44.4762 | -73.2106 | justB |
| VT | 500210002 | 43.60806 | -72.9828 | justB |

| | | | | |
|---|---|---|---|---|
| East Chicago, IN | 180895503 | 41.64921 | -87.4475 | PB |
| Gary, IN | 180895504 | 41.5997 | -87.3443 | PB |
| IN | 180190009 | 38.27668 | -85.7638 | justB |
| IN | 180855502 | 41.23898 | -85.8321 | justB |
| IN | 180890022 | 41.60668 | -87.3047 | justB |
| IN | 180890023 | 41.65274 | -87.4396 | justB |
| IN | 180890030 | 41.6814 | -87.4947 | justB |
| IN | 180970078 | 39.8111 | -86.1145 | justB |
| IN | 180970084 | 39.75885 | -86.1154 | justB |
| IN | 181270024 | 41.61756 | -87.1993 | justB |
| IN | 181570008 | 40.43164 | -86.8525 | justB |
| IN | 181630016 | 37.97444 | -87.5323 | justB |
| Follansbee, WV | 540095501 | 40.33564 | -80.5953 | PB |
| WV | 540390010 | 38.3456 | -81.6283 | justB |
| WV | 540610003 | 39.64937 | -79.9209 | justB |
| WV | 540690010 | 40.11488 | -80.701 | justB |
| Richmond, VA | 510870014 | 37.55655 | -77.4004 | PB |
| VA | 510330001 | 38.20087 | -77.3774 | justB |
| VA | 510590030 | 38.77335 | -77.1047 | justB |
| VA | 516700010 | 37.28962 | -77.2918 | justB |
| VA | 518100008 | 36.84188 | -76.1812 | justB |
| OH | 391450020 | 38.60934 | -82.82251 | N/A |
| OH | 391450021 | 38.60066 | -82.82964 | N/A |

**Appendix B: Gas-particle partitioning analysis**

[revised manuscript text omitted]

---

## Referee Comment (RC2) · Anonymous Referee #2 · 18 Jun 2018

General comment The present manuscript describes the improvements in PAH modelling in North America. The results with an analysis and discussion of the biases are lengthy and clearly presented. The strength and remaining limitations of the modelling system are put in evidence.

Major comment Possible reasons like the missing reaction with NO3 radicals are given for the high BaP model bias. I would like to see a discussion of what the recent results of Mu et al. 2018(DOI:10.1126/sciadv.aap7314), if implemented in GEM-MACH-PAH, would change the predictions for BaP. The reduced OA diffusivity would increase BaP lifetime especially in winter. On the other hand the ROI temperature-dependent reac-

tion of BaP is predicted to be the major cause of changes compared to the Kwamena's parameterization.

Minor comments In a few instances references to figure panels a), b) and c) are given although no trace of it can be found on figure 6 and 7, for example.

———————————————

---

## Author Comment (AC2) · 21 Jun 2018

Italic font is quote from referee.
Responses start with [CW]

*General comment: The present manuscript describes the improvements in PAH modelling in North America. The results with an analysis and discussion of the biases are lengthy and clearly presented. The strength and remaining limitations of the modelling system are put in evidence.*

[CW] Thank you for your review of our paper. Below we will address your comments.

[Figure]

*Major comment: Possible reasons like the missing reaction with NO3 radicals are given for the high BaP model bias. I would like to see a discussion of what the recent results of Mu et al. 2018(DOI:10.1126/sciadv.aap7314), if implemented in GEM-MACH-PAH, would change the predictions for BaP. The reduced OA diffusivity would increase BaP lifetime especially in winter. On the other hand the ROI temperature-dependent reaction of BaP is predicted to be the major cause of changes compared to the Kwamena's parameterization.*

[CW] Thank you for pointing us to this recent paper by Mu et al. As noted in the original manuscript (p8, line 257), we chose the Kwamena approach since the BaP-O3 scheme from that work were mid-range of the three available in the literature (including Pöschl and Kahan as well), while that described in Mu et al presents a fourth option, potentially worth considering for low temperature conditions (e.g., in Arctic or global simulations). We have added the following discussion to our revised paper:

"Additionally, Mu et al (2018) suggest that the heterogeneous BaP-$O_3$ reaction should be temperature-, humidity-, and organic aerosol phase state-dependent (none of which are taken into account in the Kwamena scheme used in our work). However, it has been shown that the Kwamena scheme and the Mu scheme produce similar results in mid-latitudes (where our study is located) (Mu et al, 2018). Spring/summertime BaP would be minimally affected, as outdoor temperatures at that time of year resemble the room temperature laboratory conditions that the Kwamena scheme was based on. Additionally, our positive model bias would likely increase in the fall-wintertime, when low temperatures and humidity would increase BaP lifetime in the Mu scheme."

Furthermore, we expect the relative change in results to be small with respect to the impact of emissions uncertainties.

*Minor comments: In a few instances references to figure panels a), b) and c) are given although no trace of it can be found on figure 6 and 7, for example.*

[CW] Thank you for catching those errors. In our revised manuscript, we have clarified

all figures so that (a), (b), (c), etc are included and consistent with the text and captions (Figs, 2, 6, 7, and 11 updated).
* * *